# In vivo metabolic tagging and targeting of circulating red blood cells

Yusheng Liu[1], Yizun Wang [2], Kyungwon Ko [1], Yuan Liu[1], Haiyi Huang[3], Yueji Wang [1,4], Jiadiao Zhou[1], Dhyanesh Baskaran[1], Joonsu Han[1], Rimsha Bhatta[1], Daniel Nguyen[1,2], Cecilia Leal [1,2,5,6,7], Matthew R. Berry[8], Fan Lam [2,5,6,9,10,11] & Hua Wang [1,2,5,6,7,10,11] ✉

Engineering red blood cells (RBCs) has been widely explored for drug delivery, imaging, vaccination, and other applications. However, effective strategies to directly engineer RBCs in vivo are still lacking. Here, we report successful metabolic glycan labeling of RBCs in vivo. We demonstrate that systemically administered azido-sugars can metabolically label circulating RBCs with azido groups, through labeling of both mature RBCs and RBC precursor cells. The surface azido tags on RBCs can persist for over 42 days in female mice (nearly the lifespan of RBCs), while tags on leukocytes decay to negligible levels within 3 days. Azido-labeled RBCs can covalently capture dibenzocyclooctyne-bearing cargos in vivo via click chemistry, extending cargo circulation from hours to over 35 days. This RBC tagging and targeting technology can improve fluorescence imaging of blood vessels, enable long-term MRI of brain vasculatures with a single gadolinium dose, and improve the pharmacokinetics of drugs.

Red blood cells (RBCs) constitute 40–45% of whole blood volume and >99% of blood cells, and are responsible for transporting oxygen to all tissues[1,2]. RBC dysfunction is often associated with severe diseases and life-threatening complications, demanding compatible and effective tools to probe and modify aberrant RBCs[3–5]. Considering their long life span (-120 days in humans and ~45 days in mice) and ability to access deep tissues[6,7], RBCs are also an attractive engineering target for drug delivery, imaging, hemostasis, vaccination, and other applications[8–13]. For example, drugs bounded to RBCs could exhibit improved pharmacokinetics and tissue accumulation, potentially achieving better therapeutic efficacy with a reduced dose of drugs[14–16]. Contrast agents conjugated to circulating RBCs also enable long-term imaging of vasculatures and tissues. However, a safe and effective method to directly modify or engineer circulating RBCs in vivo still does not exist. Existing RBC engineering approaches mostly rely on isolating and modulating RBCs under ex vivo conditions, and the whole process of isolating RBCs, modifying RBCs ex vivo, and infusing engineered RBCs is time-consuming and costly, could cause damage to RBCs, and suffers from a risk of infections[9,17–19]. Significant progress has indeed been made to address these issues and bring RBC-based drug delivery systems to clinical trials[20]. In parallel, we envision direct in vivo engineering of RBCs will provide possibilities for better understanding of RBC biology and developing enhanced diagnostics and therapies for a variety of diseases.

[1]Department of Materials Science and Engineering, University of Illinois at Urbana-Champaign, Urbana, IL, USA. [2]Department of Bioengineering, University of Illinois at Urbana-Champaign, Urbana, IL, USA. [3]Department of Chemistry, University of Illinois at Urbana-Champaign, Urbana, IL, USA. [4]Department of Mechanical Science and Engineering,, University of Illinois at Urbana-Champaign, Urbana, IL, USA. [5]Carle College of Medicine, University of Illinois at Urbana-Champaign, Urbana, IL, USA. [6]Beckman Institute for Advanced Science and Technology, University of Illinois at Urbana-Champaign, Urbana, IL, USA. [7]Materials Research Laboratory, University of Illinois at Urbana-Champaign, Urbana, IL, USA. [8]Department of Veterinary Clinical Medicine, University of Illinois at Urbana-Champaign, Urbana, IL, USA. [9]Department of Electrical and Computer Engineering, University of Illinois at Urbana-Champaign, Urbana, IL, USA. [10]Cancer Center at Illinois (CCIL), Urbana, IL, USA. [11]Institute for Genomic Biology, University of Illinois at Urbana-Champaign, Urbana, IL, USA. ✉e-mail: huawang3@illinois.edu

Modification of RBCs was attempted via non-covalent or covalent methods. Non-covalent methods involve physical adsorption of cargos onto RBC membranes via electrostatic interactions[11,21,22] or binding of a targeting ligand to RBC receptors (e.g., Ter119)[13,23–25]. The former is limited by the weak interactions between cargos and RBCs and the need of custom-designed adsorbable structures[11,21,22], while the latter is limited by the lack of specificity and low abundance of RBC receptors[13,23–25]. The covalent conjugation of cargos to RBCs via amine-carboxyl chemistry was also attempted, but the density of amine groups on RBC membranes is minimal, and the modified RBCs often exhibit impaired function and viability[26,27]. Another covalent method involves the genetic engineering of cell-surface proteins with an LPXTG motif, followed by the conjugation of (Gly)$_n$-bearing molecules in the presence of sortase[19,24,28]. However, the necessity of introducing the sortase-cleavable LPXTG motif via gene transduction poses a notable drawback, especially for in vivo applications. To date, a facile, safe, and universal method for in vivo engineering and cargo attachment of RBCs still does not exist.

Here we report a facile and safe strategy to metabolically label RBCs with unique chemical tags (e.g., azido groups) in vivo (Fig. 1a). We show that unnatural sugars such as tetraacetyl-*N*-azidoacetylmannosamine (AAM), upon systemic administrations, can metabolically label RBCs with azido groups in the form of glycoproteins and glycolipids via the metabolic glycoengineering processes[29–32]. The type, dose, and dosing frequency of azido-sugars all impact the overall RBC labeling efficiency. It is noteworthy that non-specific azido-sugars used in this study can label both RBCs and non-RBCs in vivo. However, due to the

non-proliferative property and long life-span of RBCs, the installed azido groups are stable in the bloodstream for over 42 days, nearly the life-span (~45 days) of mouse RBCs. In contrast, the azido tags on the membrane of leukocytes in the blood and cells in healthy tissues such as liver, spleen, lung, heart, brain, and kidney decay to baseline levels within 3 days (Fig. 1a). We further demonstrate that the azido-labeled RBCs can rapidly capture dibenzocyclooctyne (DBCO)-bearing cargos in the bloodstream via efficient click chemistry[33–37]. Strikingly, the conjugated cargos can retain on the surface of RBCs for over 35 days in mice (Fig. 1a), without causing any noticeable acute or chronic toxicity. This could be attributed to the unique physiology of RBCs and opens up opportunities for various diagnostic and therapeutic applications. As a few examples, we demonstrate that the RBC labeling and targeting technology enables in vivo conjugation of fluorophores onto circulating RBCs for enhanced fluorescence imaging of vasculatures and tumors (Fig. 1b), conjugation of gadolinium (Gd) onto RBCs for long-term magnetic resonance imaging (MRI) of brain blood vessels (for over 11 days with a single dose of Gd) (Fig. 1c), and conjugation of insulin and other drugs for enhanced pharmacokinetics and therapeutic efficacy (Fig. 1d).

## Results
### In vitro metabolic labeling of RBCs
We first screened the metabolic labeling efficiency of three types of azido-sugars, AAM, tetraacetyl-*N*-azidoacetylgalactosamine (AAGal), and tetraacetyl-*N*-azidoacetylglucosamine (AAGlu), using murine erythroleukemia (MEL) cells, a precursor cell of RBCs. MEL cells were

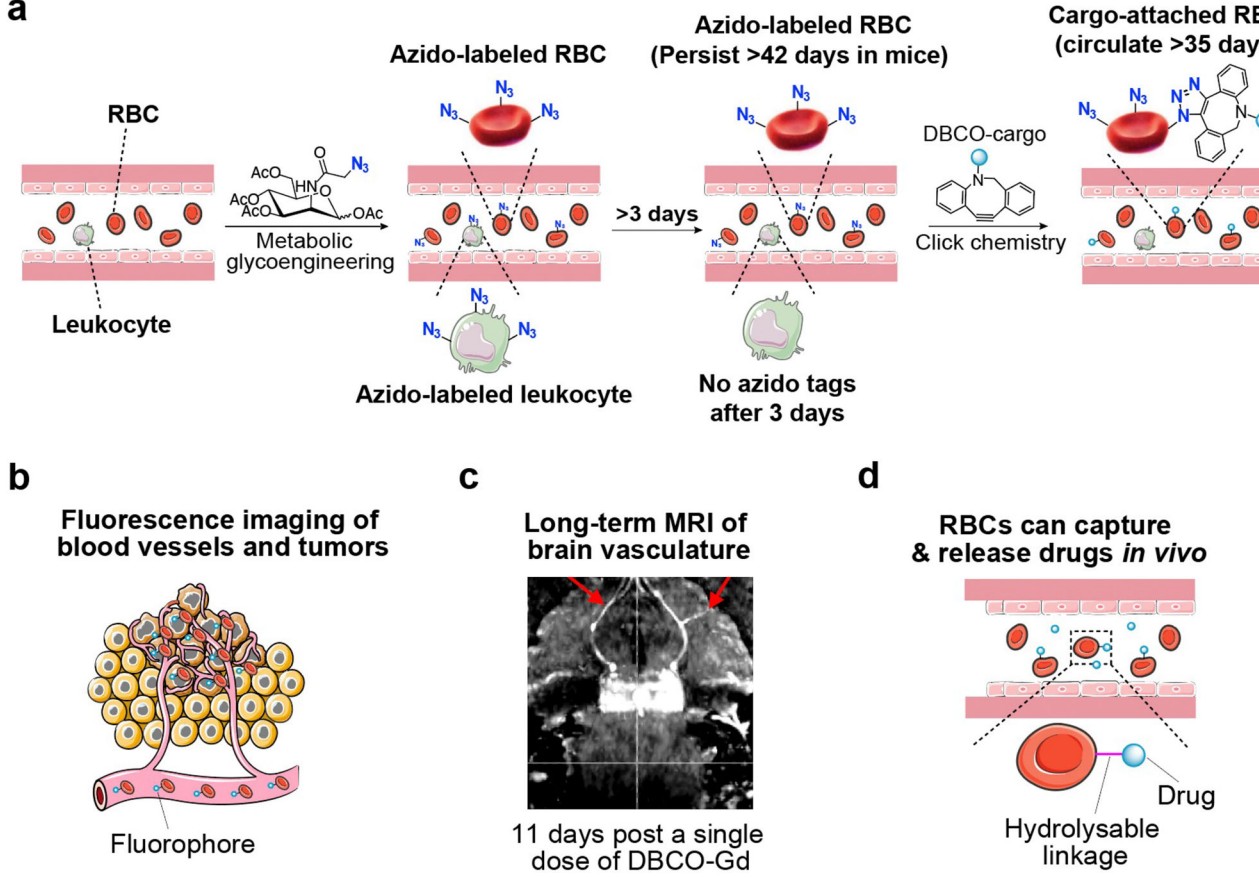

**Fig. 1 | In vivo RBC labeling and targeting. a** In vivo metabolic glycan labeling of RBCs with azido groups for subsequent conjugation of DBCO-cargo via efficient click chemistry. Azido tags on nucleus-free RBCs can persist for >42 days (nearly life-span of mouse RBCs) while azido tags on DNA-containing cells become undetectable after 3 days, enabling specific conjugation of DBCO-cargos to circulating RBCs at 3 days or later. **b** In vivo conjugation of fluorophores to RBCs for fluorescence imaging of blood vessels and tumors. **c** In vivo conjugation of Gd to RBCs for long-term MRI of brain vasculatures. **d** In vivo conjugation of drugs (e.g., insulin) to RBCs for improved pharmacokinetics and therapeutic efficacy.

incubated with 50 μM AAM, AAGal, or AAGlu for varied times, followed by the detection of cell-surface azido groups using DBCO-Cy5. Compared to control cells without azido-sugar treatment, cells treated with AAM, AAGal, or AAGlu showed significantly higher Cy5 fluorescence intensity (Supplementary Fig. 1a–c), demonstrating the successful labeling of MEL cells with azido groups. AAM exhibited a higher labeling efficiency than AAGal and AAGlu (Supplementary Fig. 1a, b). As MEL cells can proliferate rapidly, the amount of cell-surface azido groups decreased from 24 h to 48 h (Supplementary Fig. 1d–f). By adding fresh AAM to the media at 48 h, the density of cell-surface azido groups could be further increased (Supplementary Fig. 1d–f). It is noteworthy that AAM treatment induced negligible changes in the viability and differentiation of MEL cells (Supplementary Fig. 1g, h).

We next studied whether AAM can metabolically label primary RBCs in vitro (Supplementary Fig. 2a). As conventional Alsever's solution and FBS-containing DMEM media failed to maintain the long-term survival of primary RBCs, we tested different combinations of medium components and narrowed down to a type of DMEM medium containing 10% FBS, 5% BSA, non-essential amino acids, and 10 g/L glucose, which was able to maintain the high viability of RBCs isolated from C57BL/6 mice for up to 120 h in vitro. Mouse RBCs were incubated with AAM for 24 h, and the cell-surface azido groups were detected by DBCO-Cy5. Compared to control cells, AAM-treated RBCs showed significantly higher Cy5 fluorescence intensity (Supplementary Fig. 2b, c), indicating the successful labeling of RBCs with azido groups. We also studied whether AAM treatment could induce the apoptosis of RBCs by examining the expression level of phosphatidylserine, which showed negligible differences between AAM- and PBS-treated RBCs (Supplementary Fig. 2d, e). These experiments demonstrated that AAM can metabolically label primary RBCs with azido groups in vitro, without inducing the apoptosis or death of RBCs.

## In vivo metabolic labeling of RBCs
We next studied whether systemically administered AAM can metabolically label RBCs with azido groups in vivo (Fig. 2a). C57BL/6 mice were intravenously administered with AAM (100 mg/kg) twice daily for three days, followed by the collection of RBCs at different times for azido detection via DBCO-Cy5. Compared to control RBCs that showed minimal Cy5 signal, RBCs collected from AAM-treated mice at 14 days post AAM injections showed a distinct Cy5$^+$ population in FACS assay (Fig. 2b), indicating the successful metabolic labeling of RBCs with azido groups. Fluorescence imaging also confirmed the much higher Cy5 fluorescence intensity of RBCs from AAM-treated mice (Fig. 2c, d). The percentage of azide$^+$ RBCs increased from 4.2% at 24 h to 10.4% at 5 days, and further to 11.3% at 7 days (Fig. 2e). While the percentage of azide$^+$ RBCs started to decrease from day 7, a notable fraction of azide$^+$ RBCs could be consistently detected for over 42 days (Fig. 2e, f), nearly the life-span of mouse RBCs (~45 days). The Cy5 fluorescence intensity ratio of RBCs (AAM/PBS) constantly stayed above 1.0, with a maximal ratio of 3.1 at 7 days (Fig. 2g). These experiments demonstrated the successful metabolic labeling of RBCs with azido groups in vivo and the superior retention of azido groups on the surface of RBCs. It was not surprising to detect azido-labeled immune cells and other cells in the bloodstream at 24 h post the injections of AAM (Fig. 2h, i). However, due to their high proliferation rate and active metabolisms, the number of azido groups decayed to the baseline level after 3 days (Fig. 2h, i). The number of azide$^+$ RBCs is 28-fold, 93-fold, 583-fold, and 3844-fold of azido$^+$ white blood cells (WBCs) at 0.5, 2.5, 4.5, and 7.5 days post the injections of AAM (Fig. 2j, k, Supplementary Fig. 3a). It is noteworthy that AAM treatment did not alter the percentage and number of RBCs and WBCs in the bloodstream (Supplementary Fig. 3b).

To further confirm the successful metabolic labeling of RBCs, we performed magnetic selection of azido-positive RBCs collected from C57BL/6 mice at 7 days post injections of AAM via DBCO-desthiobiotin

conjugation, streptavidin-microbead pull-down, and biotin elution (Fig. 2l). Flow cytometry analyses confirmed the successful enrichment of desthiobiotin-tagged RBCs (Fig. 2m, n and Supplementary Fig. 3c). Western blot analysis of the enriched azido/desthiobiotin-tagged RBCs versus control azido$^-$ RBCs was able to identify several types of azido-labeled proteins such as band 3 proteins (100 kDa), bands 4 and 5 proteins (55–60 kDa), and glycophorins A–D (30–35 kDa) (Fig. 2o). It is noteworthy that western blot analysis of RBCs, without magnetic enrichment, also showed a difference between AAM and PBS groups (Supplementary Fig. 3d, e).

## Mechanisms underlying in vivo metabolic labeling of RBCs
We further studied whether azido-labeled glycoproteins are primarily expressed on RBC membrane by conjugating RBCs from AAM-treated mice with DBCO-desthiobiotin, isolating the membrane and cytoplasmic proteins, and performing western blot analysis (Fig. 3a). The cytoplasmic proteins of RBCs in both AAM and PBS groups showed minimal desthiobiotin signal (Fig. 3b). In comparison, the membrane proteins of RBCs in AAM group showed significantly higher desthiobiotin signals (Fig. 3b). These results demonstrated that azido-labeled glycoproteins mainly exist on the outer membrane instead of in the cytoplasm. To further validate the presence of azido-labeled glycoproteins, the membrane proteins of desthiobiotin-conjugated RBCs were treated with PNGase F, an enzyme catalyzing the removal of N-linked oligosaccharide chains from glycoproteins, for 1 h prior to western blot analysis. As a result, PNGase F treatment eliminated nearly all the desthiobiotin signal from proteins (Fig. 3c), validating the metabolic expression of azido groups in the form of glycoproteins in azido-labeled RBCs. We also tested the presence of azido-tagged glycolipids on the membrane of RBCs isolated from AAM-treated mice, by incubating RBCs with DBCO-Cy5, extracting membrane lipids from RBCs, and analyzed with HPLC under the fluorescence mode. Lipids isolated from AAM-treated 4T1 cancer cells were used as the positive control. As a result, HPLC analyses confirmed the presence of Cy5-tagged lipids in AAM-treated RBCs and 4T1 cells instead of PBS-treated RBCs (Fig. 3d and Supplementary Fig. 3f).

We next studied whether metabolic labeling of RBC precursor cells in the bone marrow contributes to the overall RBC labeling efficiency. Mouse bone marrow cells were harvested at 48 h after AAM injections and stained with DBCO-Cy5 to analyze azido-labeled RBC precursor cells including proerythroblasts, basophilic erythroblasts, polychromatophilic erythroblasts, orthochromatophilic erythroblasts, and reticulocytes (Supplementary Fig. 4a). Compared to the PBS group, basophilic erythroblasts, polychromatophilic erythroblasts, orthochromatophilic erythroblasts, and reticulocytes in the AAM group showed a higher Cy5 fluorescence intensity (Fig. 3e). The ratio of Cy5 mean fluorescence intensity (AAM/PBS) increased with the stage of erythropoiesis (Fig. 3f). These results demonstrate that systemically administered AAM can metabolically label RBC precursor cells in the bone marrow, which could contribute to the overall RBC labeling efficiency. It is noteworthy that AAM treatment did not alter the percentage and number of erythroid precursor cells (Supplementary Fig. 4b, c).

We also assessed and compared in vivo RBC labeling efficiency of AAM, AAGal, and AAGlu. C57BL/6 mice were intravenously injected with AAM, AAGal, AAGlu, or PBS twice daily for three days, and RBCs were collected at different times and stained with DBCO-Cy5 for flow cytometry analysis. Compared to AAGlu, AAM and AAGal showed a significantly higher RBC labeling efficiency (Fig. 3g, h). Compared to the AAGal group, azido-labeled RBCs in AAM-treated mice exhibited a higher stability over time (Fig. 3g, h). In addition to intravenous injections, we showed that intraperitoneal injections of AAM, which allowed for the use of a higher dose, also resulted in successful metabolic labeling of RBCs with azido groups (Supplementary Fig. 5a–d). The RBC metabolic labeling efficiency increased with the

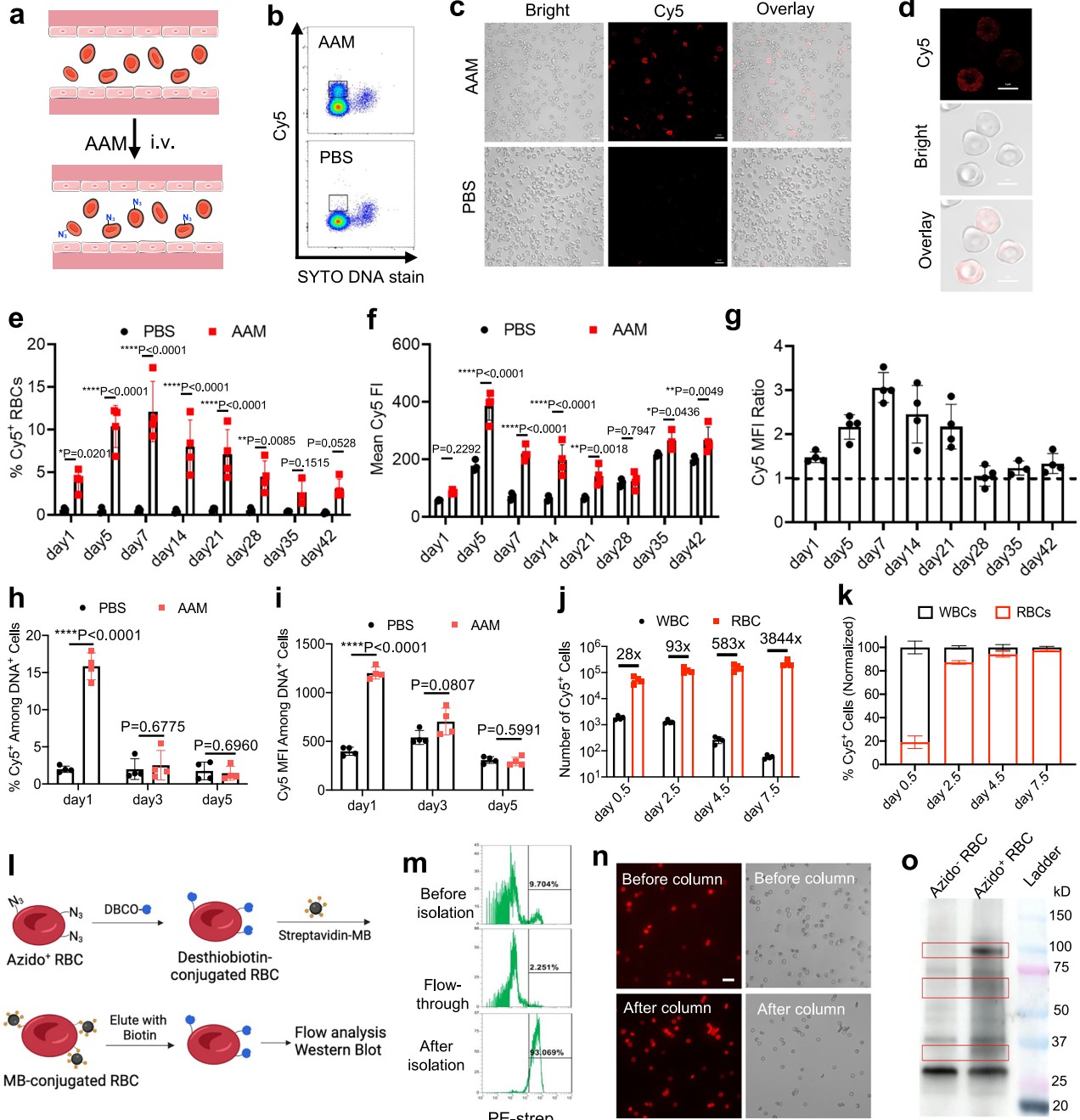

**Fig. 2 | Systemically administered azido-sugar can metabolically label RBCs with azido groups in vivo and azido groups persist on circulating RBCs for >42 days in mice. a** Schematic illustration of in vivo metabolic labeling of circulating RBCs. **b–k** AAM was intravenously injected into C57BL/6 mice twice daily for three days. RBCs isolated from mice were incubated with DBCO-Cy5 and nucleic acid stain (SYTO) for 1 h. **b** Representative flow cytometry plot of RBCs at 14 days post AAM injections. **c** Fluorescence images and **d** confocal images of RBCs harvested at 14 days post AAM injections. Also shown are **e** % Cy5+ RBCs, **f** mean Cy5 fluorescence intensity of RBCs, and **g** Cy5 fluorescence intensity ratio of RBCs (AAM/PBS) harvested at different times post AAM injections and stained with DBCO-Cy5 ($n = 4$ mice per group). **h** % Cy5+ cells among DNA+ cells and **i** mean Cy5 fluorescence intensity of DNA+ cells in the blood at 1, 3, and 5 days, respectively, post AAM injections ($n = 4$ mice per group). **j** Number of Cy5+ white blood cells (WBCs) and RBCs in the blood at day 0.5, 2.5, 4.5, and 7.5, respectively, post AAM injections ($n = 4$ mice per group). **k** Normalized percentages of Cy5+ WBCs and

RBCs in the blood at day 0.5, 2.5, 4.5, and 7.5, respectively, post AAM injections ($n = 4$ mice per group). **l** Schematic illustration of purification of azido-labeled RBCs, by conjugating DBCO-desthiobiotin to RBCs and pulling down RBCs using streptavidin-microbeads. The figure was created with BioRender. **m** Representative PE-streptavidin histograms of RBCs before and after purification. **n** Fluorescence images of RBCs before and after purification. Scale bar: 20 μm. **o** Western blot analysis of proteins extracted from purified azido+ RBCs and azido- RBCs. The protein bands/area that are enriched in azido+ RBCs are indicated by red rectangles. All the numerical data are presented as mean ± SD. Statistical significance in (**e**, **f**) was determined by ordinary two-way ANOVA followed by Fisher's LSD test using a pooled variance. For (**h**) and (**i**), multiple unpaired $t$-tests were performed with Holm-Sidak correction for multiple comparisons ($0.01 < *P \leq 0.05$; $**P \leq 0.01$; $***P \leq 0.001$; $****P \leq 0.0001$). All experiments were performed at least three times independently with similar results. Source data for this figure is available in the Source data file.

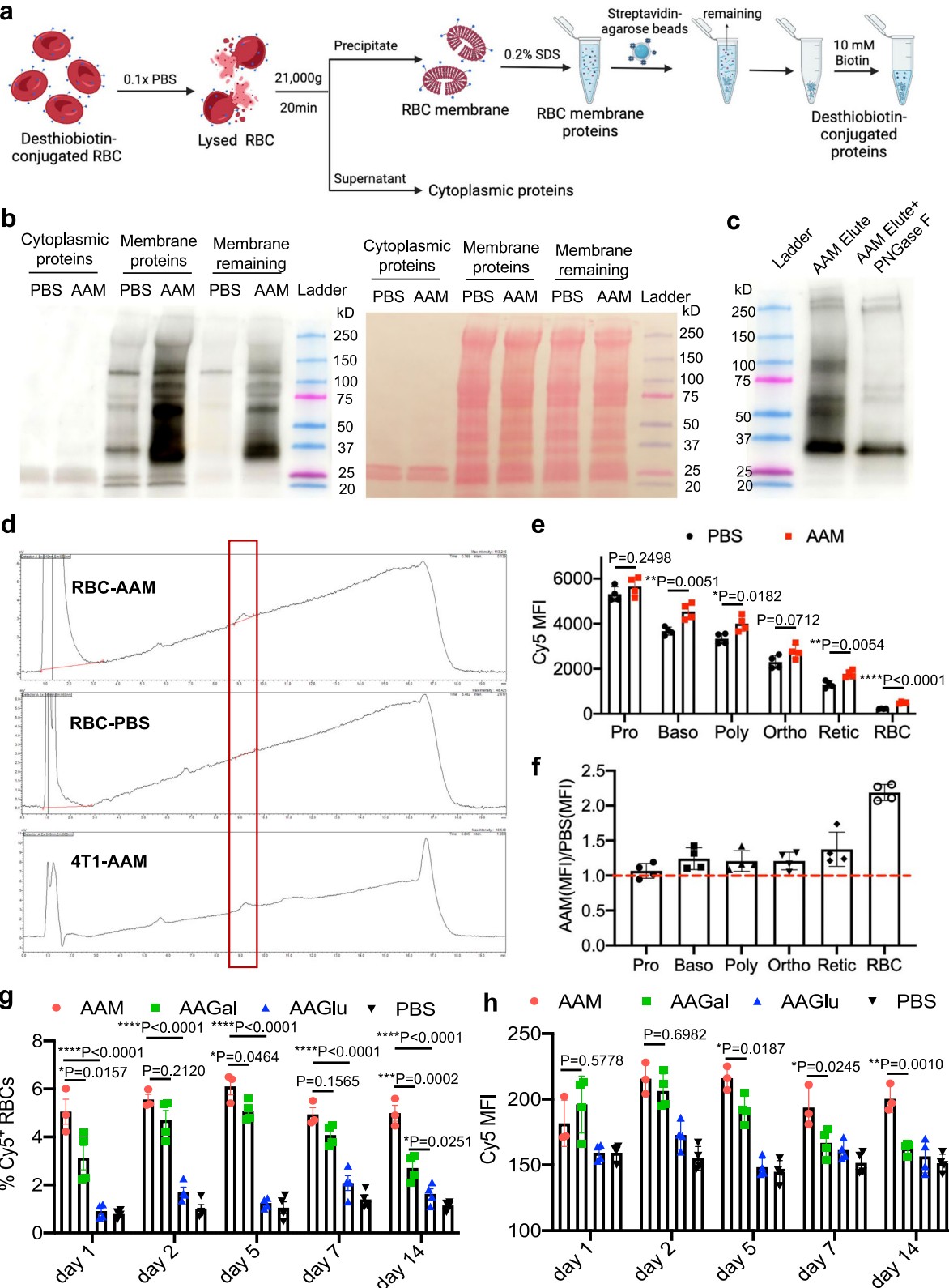

dosing frequency of AAM, with up to 14.5% of azido-labeled RBCs among all the circulating RBCs (Supplementary Fig. 5a–d).

**In vivo metabolic labeling of RBCs does not induce noticeable toxicity**

We also studied whether in vivo metabolic labeling of RBCs could pose any toxic effects on blood cells and tissues. RBCs from AAM- or PBS-treated mice both showed the normal biconcave disc morphology and

exhibited minimal agglutination (Fig. 4a), which was further confirmed by flow cytometry (Fig. 4b). As a key indicator of RBC metabolic states, the intracellular ATP level was measured and compared between AAM and PBS groups. Per the ATP bioluminescence assay, RBCs from AAM- or PBS-treated mice showed negligible differences in intracellular ATP levels (Fig. 4c). In addition to ATP, we also measured the intracellular NAD(H)/NADP(H) levels of RBCs using the resazurin assay. Compared to control RBCs, RBCs in AAM-treated mice showed negligible changes

**Fig. 3 | Azido-sugar can metabolically label surface glycoproteins and glycolipids of RBCs in the blood and RBC precursor cells in the bone marrow.**
**a** Schematic illustration for the isolation of membrane proteins and cytoplasmic proteins from RBCs that were collected at 7 days post injections of AAM or PBS. RBCs were incubated with DBCO-dethiobiotin first, prior to the isolation of membrane and cytoplasmic proteins and western blot analysis. Figure was created with BioRender. **b** Western blot analysis of dethiobiotin-tagged proteins in the cytoplasmic fraction, membrane fraction, and remaining membrane fraction (supernatant after streptavidin-agarose bead pull-down). Dethiobiotin signal was detected using streptavidin-HRP. **c** Western blot analysis of dethiobiotin-tagged proteins in the membrane fraction with or without PNGase F treatment. (d) Representative HPLC profile of lipids from RBCs at 7 days post injections of AAM or PBS. RBCs isolated from AAM- or PBS-treated mice were incubated with DBCO-Cy5, followed by lipid extraction and HPLC analysis ($\lambda_{ex}$: 645 nm, $\lambda_{em}$: 660 nm). 4T1 cells treated with AAM for 24 h in vitro were used as controls. **e, f** Bone marrow cells were

collected at 48 h post injections of AAM or PBS, incubated with DBCO-Cy5, and stained for erythroid lineage markers prior to follow cytometry analysis. Shown are **e** mean Cy5 fluorescence intensity (FI) and **f** Cy5 FI ratio (AAM/PBS) of different RBC precursor cells ($n = 4$ mice per group). Pro: proerythroblast. Baso: Basophilic erythroblast. Poly: Polychromatophilic erythroblast. Ortho: Orthochromatophilic erythroblast. Retic: Reticulocyte. C57BL/6 mice were i.v. injected with AAM, AAGal, AAGlu, or PBS twice daily for two days. RBCs were collected at different times and stained with DBCO-Cy5. Shown are (**g**) % Cy5$^+$ RBCs and (**h**) mean Cy5 FI of RBCs at 1, 2, 5, 7, and 14 days, respectively ($n = 4$ mice per group). All the numerical data are presented as mean ± SD. Multiple unpaired $t$-tests were performed for (**e**), and one-way ANOVA with Turkey test for multiple comparisons was used for (**g**, **h**) ($0.01 < *P \le 0.05$; $**P \le 0.01$; $***P \le 0.001$; $****P \le 0.0001$). All experiments were performed at least three times independently with similar results. Source data for this figure is available in the Source data file.

in NAD(H)/NADP(H) levels, both showing a much higher NAD(H)/NADP(H) level than the negative control (Fig. 4d). The RBC and WBC counts showed no differences between the AAM and PBS groups (Fig. 4e, f). Different subtypes of WBCs, including T cells, B cells, and granulocytes, also showed no differences between AAM and PBS groups (Fig. 4g). To study the potential chronic toxicity, organs were harvested at 6 weeks post AAM injections, sectioned, and stained with H&E. Compared to the PBS group, AAM treatment did not induce any noticeable toxicity against all the examined tissues including liver, spleen, kidney, heart, and lung (Fig. 4h). We also analyzed the potential metabolic labeling of cells in healthy tissues. At 4 days post injections of AAM or PBS, tissues were homogenized, and the extracted proteins were incubated with DBCO-biotin prior to western blot analysis. Tissues, including brain, liver, spleen, kidney, heart, and lung showed negligible differences between AAM and PBS groups (Fig. 4i). Confocal imaging of tissue sections, after staining with DBCO-Cy5, also showed negligible differences in Cy5 fluorescence intensity between AAM and PBS groups (Fig. 4j, k). We also analyzed the membrane osmotic and mechanical fragility of RBCs collected from AAM-treated mice (both azide-positive and azide-negative fractions) and PBS-treated mice, which showed negligible differences (Supplementary Fig. 6a, b). RBC indices, including RBC counts, Hemoglobin concentration, hematocrit percentages, mean cell volume (MCV), mean corpuscular hemoglobin (MCH), mean corpuscular hemoglobin concentration (MCHC), and red cell distribution width (RDW) (Supplementary Fig. 7a–g), as well as RBC morphology (Supplementary Fig. 7h), showed negligible differences between AAM and PBS groups. These data demonstrated that AAM-mediated metabolic labeling induced negligible changes in RBC membrane fragility and RBC functions.

## In vivo targeting of DBCO-cargo to azido-labeled RBCs
We next studied whether circulating azido-labeled RBCs would enable targeted conjugation of DBCO-molecules via click chemistry in vivo. C57BL/6 mice were intravenously injected with AAM or PBS twice daily for three days, followed by intravenous injection of DBCO-Cy5 at three days post the last injection of AAM (Fig. 5a). At 3 h post injection of DBCO-Cy5, 3.7% of RBCs in AAM-treated mice were Cy5-positive, which was much higher than 0.8% in PBS-treated mice (Fig. 5b). Mean Cy5 fluorescence intensity of RBCs from AAM-treated mice was also significantly higher than control RBCs (Fig. 5c, d). At 9 h post injection of DBCO-Cy5, the percentage of Cy5$^+$ RBCs and mean Cy5 fluorescence intensity stayed much higher in the AAM-treated mice than in the control mice (Fig. 5b–d). After 24 or 48 h, when the vast majority of the injected DBCO-Cy5 should have been cleared from the bloodstream, a significantly higher number of Cy5$^+$ RBCs were still detected in AAM-treated mice than in control mice (Fig. 5b–d). To monitor the long-term retention of Cy5$^+$ RBCs in the bloodstream, we also harvested RBCs at 5, 7, 14, 21, 28, and 35 days respectively post the injection of DBCO-Cy5, which all showed higher numbers of Cy5$^+$ RBCs in AAM-

treated mice than in control mice (Fig. 5b–d). As expected, blood isolated from AAM-treated mice at days 1, 4, 7, and 14 also showed higher Cy5 fluorescence intensity than blood from PBS-treated mice (Fig. 5e, f). Among DNA-containing cells (non-RBCs) in the bloodstream, the accumulation of DBCO-Cy5 showed negligible differences between AAM and PBS groups at all examined times (Fig. 5g, h), demonstrating minimal off-target delivery of DBCO-Cy5 to non-RBC cells. This is consistent with the labeling results, which showed that the percentage of azide$^+$ cells among DNA-containing blood cells decayed to baseline levels at 3 days post AAM injection (Fig. 2h–k). We also examined the potential off-target accumulation of DBCO-Cy5 in healthy tissues. At 24 h post DBCO-Cy5 injection, confocal imaging of tissue sections showed negligible differences in Cy5 fluorescence intensity between AAM and PBS groups (Fig. 5i, j). To better understand the membrane retention of conjugated DBCO-Cy5 on RBCs, we also isolated the RBCs at 7 days post injections of AAM, incubated them with DBCO-Cy5, and monitored the membrane retention of Cy5 ex vivo (Supplementary Fig. 8a). As expected, at 1, 3, and 5 days post DBCO-Cy5 conjugation, RBCs from AAM-treated mice consistently showed much higher Cy5 fluorescence intensity than RBCs from PBS-treated mice (Supplementary Fig. 8b, c), substantiating the excellent membrane retention of Cy5 conjugated onto RBCs. These experiments demonstrated that azido-labeled circulating RBCs can covalently capture DBCO-molecules in vivo, and the conjugated molecules can retain on RBC membrane for >5 weeks in mice. We are aware of the concern on the potential immunogenicity of cargos being conjugated to circulating RBCs. In separate studies, we attempted to conjugate DBCO-functionalized 2,4-dinitrophenylated Hemocyanin, Keyhole Limpet (DNP-KLH), and DBCO-ovalbumin (OVA), two representative antigens, to RBCs in vivo and monitor the antibody titers over time. As a result, the in vivo conjugation of DNP-KLH and OVA antigens failed to exert any enhancement on the antibody responses compared to the control groups (Supplementary Fig. 9), which lessens the concern of potent immunogenicity of RBC-conjugated cargos in our metabolic labeling approach. We estimate <1–10% of administered DBCO-protein could be conjugated to circulating RBCs, and the in vivo RBC labeling and targeting conditions require further optimization to increase the amount of conjugated cargos.

## Long-term fluorescence imaging of blood vessels and tumors
Considering that DBCO-Cy5 can be conjugated to azido-labeled RBCs and well retain on RBCs in vivo, we next explored its promise for long-term fluorescence imaging of blood vessels and tissues. Balb/c mice were intravenously injected with AAM or PBS twice daily for three days, followed by the subcutaneous inoculation of 4T1 tumor cells and intraperitoneal injection of DBCO-Cy5 at 4 and 7 days, respectively (Fig. 6a). Starting from 1 h post injection of DBCO-Cy5, a higher Cy5 fluorescence intensity was detected in RBCs from AAM-treated mice

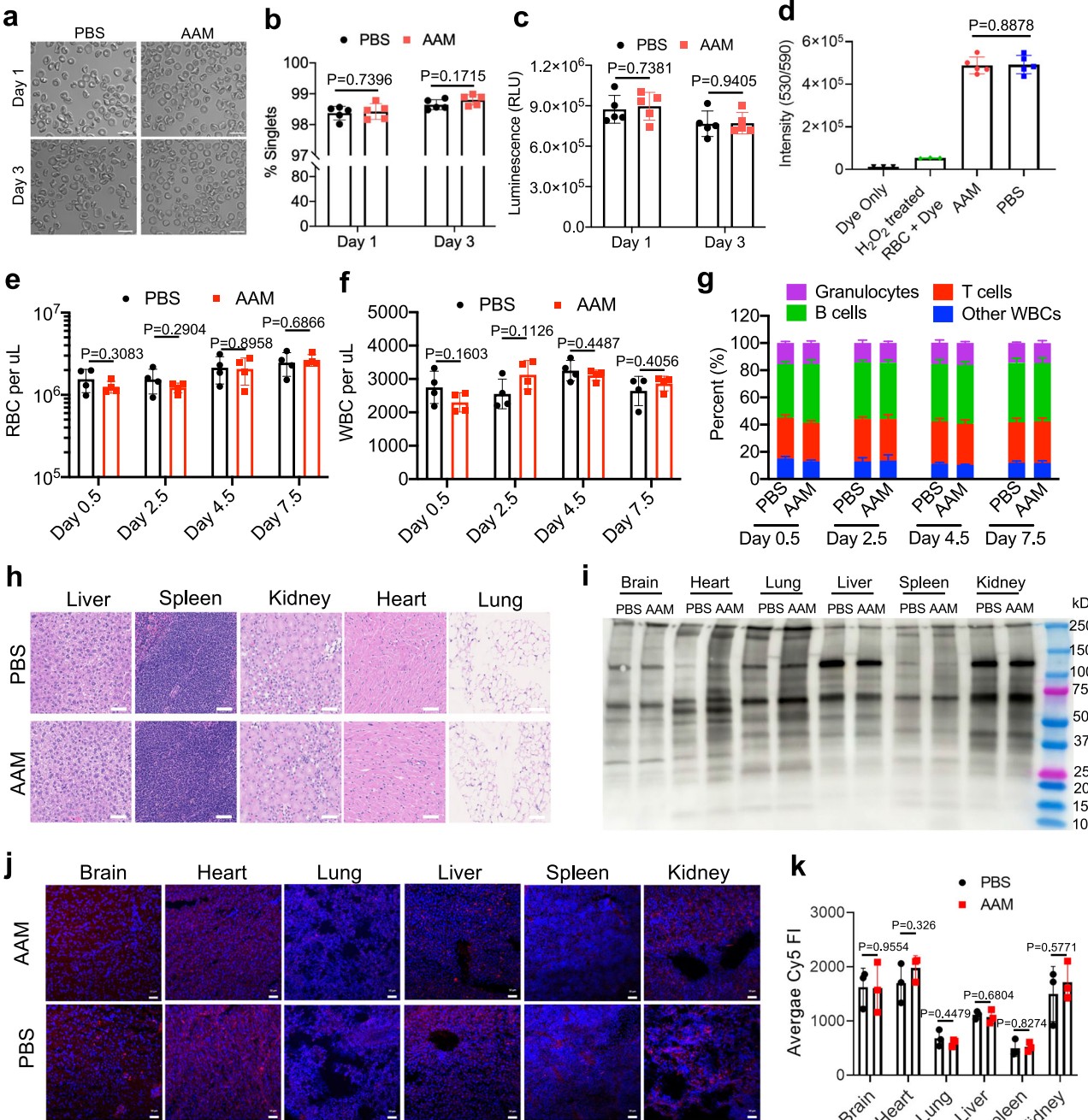

**Fig. 4 | In vivo metabolic labeling of RBC does not induce noticeable toxicity to RBCs, WBCs and healthy tissues. a** Representative microscopic images of RBCs harvested from mice at 1 or 3 days post injections of AAM or PBS. **b** % singlet RBCs harvested at 1 or 3 days post the last injection of AAM or PBS ($n = 5$ mice per group) based on flow cytometry analysis. **c** Intracellular ATP levels of RBCs collected from mice at 1 or 3 days post injections of AAM or PBS ($n = 5$ mice per group).
**d** Intracellular NAD(P)H levels of RBCs collected from mice at 3 days post injections of AAM or PBS ($n = 5$ mice per group). **e** RBC count and **f** WBC count at different times post injections of AAM or PBS ($n = 4$ mice per group). **g** % of different types of WBCs at different times post the final injection of AAM or PBS ($n = 4$ mice per group). **h** Representative images of H&E stained tissue sections at 6 weeks post

injections of AAM or PBS. Scale bar: 100 μm. **i** Western blot analysis of different organs at 4 days post AAM or PBS injections. Proteins were extracted from organs, reacted with DBCO-biotin, and run on the gel. The biotin signal was detected by streptavidin-HRP. **j** Confocal images of sectioned tissues that were collected at 4 days post injections of AAM or PBS and stained with DBCO-Cy5 and DAPI. Scale bar: 50 μm. **k** Cy5 fluorescence intensity of tissues in (**j**) ($n = 3$ mice per group). All the numerical data are presented as mean ± SD. Statistical significance was determined using multiple unpaired $t$-tests with Holm–Sidak correction for multiple comparisons (0.01 <*$P \le 0.05$; **$P \le 0.01$; ***$P \le 0.001$; ****$P \le 0.0001$). All experiments were performed at least three times independently with similar results. Source data for this figure is available in the Source data file.

than RBCs from PBS-treated mice (Fig. 6b). IVIS imaging also showed a higher Cy5 fluorescence intensity of 4T1 tumors in AAM-treated mice than in PBS-treated mice at 21 days post DBCO-Cy5 injection (Fig. 6c, d). Ex vivo imaging of 4T1 tumors confirmed a higher Cy5 fluorescence signal in the AAM group (Fig. 6e, f). Indeed, blood vessels in 4T1 tumors

of AAM-treated mice could be clearly visualized (Fig. 6e). Similarly, the blood vessels in other tissues, including the terminal arterioles structures in the spleen, could be visualized (Supplementary Fig. 10a). In another setting, C57BL/6 mice were intravenously injected with AAM or PBS twice daily for three days, followed by intravenous injection of

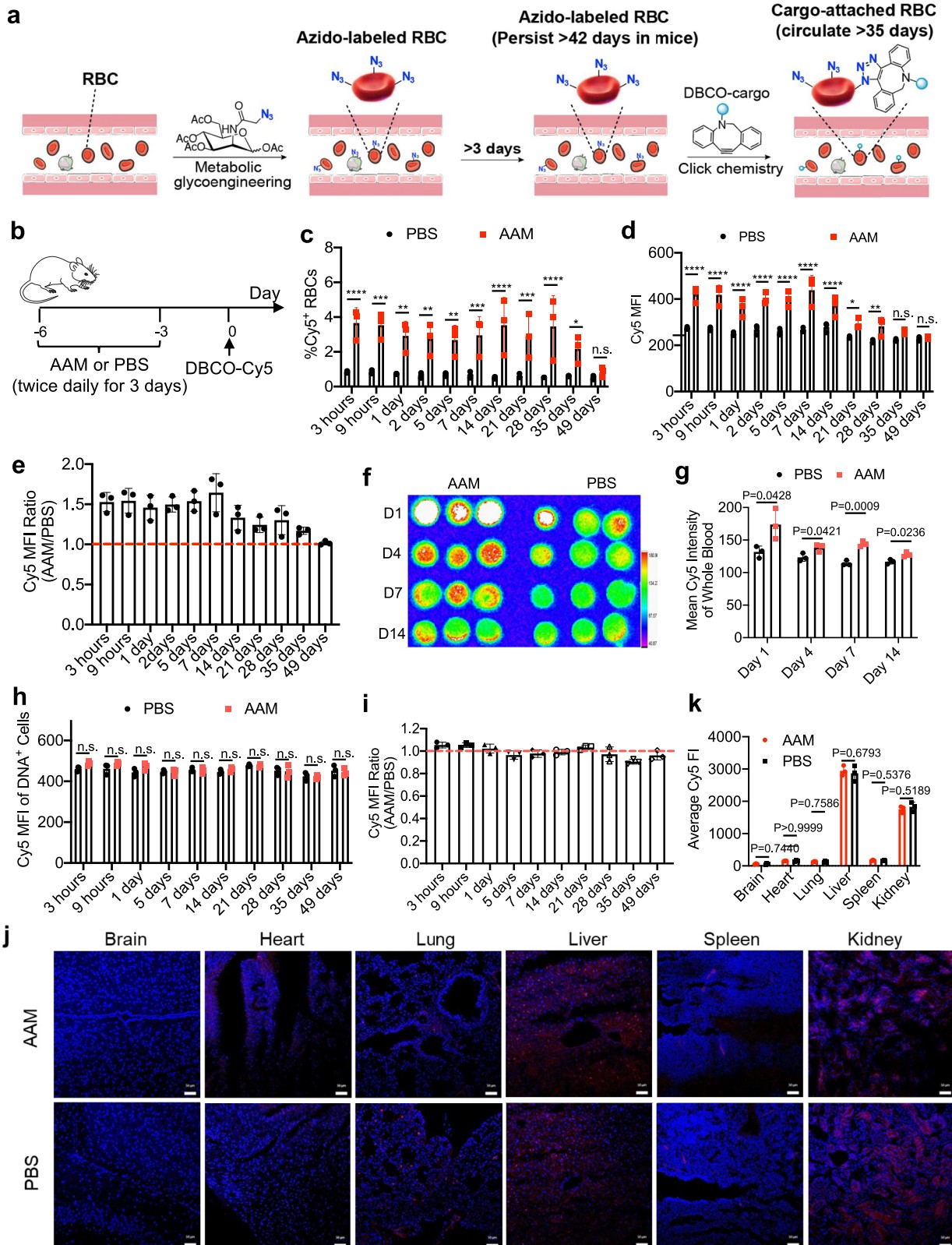

DBCO-Cy5 on day 0 and B16F10 tumor inoculation on day 17 (Supplementary Fig. 10b). On day 24, i.e., 7 days post the subcutaneous inoculation of B16F10 cells, B16F10 tumors from AAM-treated mice showed a 1.92-fold Cy5 fluorescence intensity in comparison with tumors from PBS-treated mice (Supplementary Fig. 10c, d), presumably due to the presence of Cy5-conjugated RBCs within the tumor vasculatures. Consistently, we observed a higher Cy5 fluorescence

signal in the liver, kidney, and other organs of AAM-treated mice than in those of PBS-treated mice (Supplementary Fig. 10e, f).

## MRI imaging of brain blood vessels

As cargos can be conjugated to circulating RBCs and retained on RBCs for weeks via the RBC labeling and targeting technology, we next investigated whether MRI contrast agents (e.g., Gd) can be tagged onto

**Fig. 5 | DBCO-molecules administered at 3 days post AAM injections can target azido-labeled RBCs in vivo and persist on RBC membranes for >35 days. a** In vivo metabolic glycan labeling of RBCs with azido groups for subsequent conjugation of DBCO-cargo via efficient click chemistry. Azido tags on nucleus-free RBCs can persist for >42 days (nearly life-span of mouse RBCs) while azido tags on DNA-containing cells become undetectable after 3 days, enabling specific conjugation of DBCO-cargos to circulating RBCs at 3 days or later. **b–i** in vivo RBC labeling and targeting study. AAM or PBS ($n = 3$ mice per group) was intravenously injected into C57BL/6 mice twice daily for three days, followed by intravenous injection of DBCO-Cy5 at three days after the last AAM injection. **b** Study timeline. **c** % Cy5$^+$ RBCs harvested from AAM- or PBS-treated mice at different times post injection of DBCO-Cy5. *p*-values: 3 h ($p < 0.0001$), 9 h ($p = 0.0001$), 1 day ($p = 0.0011$), 2 days ($p = 0.0017$), 5 days ($p = 0.0028$), 7 days ($p = 0.0007$), 14 days ($p < 0.0001$), 21 days ($p = 0.0006$), 28 days ($p < 0.0001$), 35 days ($p = 0.6213$), and 49 days ($p = 0.0159$). **d** Mean Cy5 fluorescence intensity of RBCs harvested from AAM- or PBS-treated mice at different times post injection of DBCO-Cy5. *p*-values: 3 h ($p < 0.0001$), 9 h ($p < 0.0001$), 1 day ($p < 0.0001$), 2 days ($p < 0.0001$), 5 days ($p < 0.0001$), 7 days ($p < 0.0001$), 14 days ($p = 0.0001$), 21 days ($p = 0.0112$), 28 days ($p = 0.004$), 35 days ($p = 0.0889$), and 49 days ($p = 0.8543$). **e** Cy5 fluorescence intensity ratio (AAM/PBS) of RBCs harvested at different times post DBCO-Cy5 injections. **f** IVIS imaging of blood ($n = 3$ mice per group) at 1, 4, 7, and 14 days post injection of DBCO-Cy5. **g** Cy5 fluorescence intensity of blood in (**f**) ($n = 3$ mice per group). **h** mean Cy5 fluorescence intensity of DNA+ cells and **i** Cy5 fluorescence intensity ratio (AAM/PBS) of DNA+ cells in the blood at different times post DBCO-Cy5 injection ($n = 3$ mice per group). **j–k** AAM or PBS was intravenously injected into C57BL/6 mice ($n = 3$ per group) twice daily for three days, followed by intravenous injection of DBCO-Cy5 at 4 days after the last AAM injection. **j** Confocal images of sectioned tissues at 24 h post injections of DBCO-Cy5. Tissue sections were stained with DAPI. Scale bar: 50 μm. **k** Cy5 fluorescence intensity of tissues in (**j**) ($n = 3$ mice per group). All the numerical data are presented as mean ± SD. Statistical significance was determined using multiple unpaired *t*-tests with Holm–Sidak correction for multiple comparisons (0.01 <*$P ≤ 0.05$; **$P ≤ 0.01$; ***$P ≤ 0.001$; ****$P ≤ 0.0001$). All experiments were performed at least three times independently with similar results. Source data for this figure is available in the Source data file.

RBCs in vivo for long-term MRI, using contrast-enhanced (CE) imaging of brain blood vessels as an example. We first synthesized DBCO-DOTA-Gd, by reacting DBCO-NH$_2$ with DOTA-COOH to yield DBCO-DOTA and furthering complexing DBCO-DOTA and Gd$^{3+}$ (Supplementary Fig. 11a). The chemical structure of DBCO-DOTA and DBCO-DOTA-Gd was characterized by $^1$H NMR spectroscopy (Supplementary Fig. 11b), HPLC (Supplementary Fig. 11c), and mass spectrometry (Supplementary Fig. 11d). For the in vivo MRI study, C57BL/6 mice were intravenously injected with AAM or PBS twice daily for three days, followed by intravenous injection of DBCO-DOTA-Gd after 4 days (Fig. 6g). Before the injection of DBCO-DOTA-Gd, the blood vessels were barely visible in a CE T1-weighted MR angiography scan (CE-MRA), with or without maximum intensity projection (MIP) (Fig. 6h). At 10 min post the injection of DBCO-DOTA-Gd, blood vessels were clearly visible in both AAM- and PBS-treated mice using the same CE-MRA sequence (Fig. 6h). At 24 h when free DBCO-DOTA-Gd was largely cleared, while MIP could visualize the main blood vessels in both AAM and PBS group, only the AAM group showed visible branched vessels (Fig. 6h), showing excellent retention of DBCO-DOTA-Gd on the azido-labeled RBCs. Without RBC labeling, Gd would need to be injected again for CE-MRA scans, as demonstrated in the PBS group. Strikingly, this signal enhancement was still detected even at 4 or 11 days post injection of DBCO-DOTA-Gd for the AAM group (Fig. 6h). To further compare the signals, we analyzed two specific coronal views of the raw MR images for both day 4 and day 11. Compared to PBS-treated mice, AAM-treated mice showed enhanced MR signals in several blood vessels on both day 4 and day 11 (Fig. 6i, k). Quantification of the signals indicated a 1.23-fold enhancement for AAM group on day 4 and a 1.13-fold enhancement on day 11 (Fig. 6j), which was consistent in different coronal views (Fig. 6j, l). These experiments demonstrated the feasibility of conjugating Gd to circulating RBCs for long-term MRI of blood vessels.

## In vivo conjugation of drugs to RBCs for improved pharmacokinetics and efficacy

In addition to imaging agents, we envision the RBC labeling and targeting approach can be utilized to improve the pharmacokinetics of therapeutics. The feasibility of conjugating small-molecule doxorubicin and macromolecular phycoerythrin (PE) onto azido-labeled RBCs was first tested. DBCO-doxorubicin with a pH-labile hydrazone linkage was synthesized (Supplementary Fig. 12a, b). Following intravenous injection of AAM or PBS into C57BL/6 mice twice daily for three days, DBCO-doxorubicin was intravenously administered (Supplementary Fig. 12c). At 2 h post the injection of DBCO-doxorubicin, a higher fraction of doxorubicin-containing RBCs was detected in AAM-treated mice than in control mice (Supplementary Fig. 12d). Similarly,

PE, upon DBCO functionalization (Supplementary Fig. 12e), can conjugate to azido-labeled RBCs via click chemistry (Supplementary Fig. 12f–h). After demonstrating the feasibility of conjugating different cargos to RBCs, we next explored whether DBCO-insulin can be conjugated to circulating RBCs for improved pharmacokinetics and therapeutic efficacy. DBCO-insulin with a hydrolysable ester linkage was synthesized and characterized (Fig. 7a and Supplementary Fig. 13a, b). By incubating AAM- or PBS-treated 4T1 cells with DBCO-insulin and detecting the cell-surface insulin using rabbit anti-insulin and Cy5-conjugated goat anti-rabbit secondary antibody, a higher Cy5 fluorescence signal was detected in AAM-treated 4T1 cells than in control cells (Supplementary Fig. 13c, d), demonstrating the successful conjugation of DBCO-insulin. DBCO-insulin was also able to conjugate to azido-labeled RBCs that were isolated from AAM-treated mice, as evidenced by antibody staining (Fig. 7b and Supplementary Fig. 13e). In vivo, following the intravenous injection of AAM or PBS into streptozotocin (STZ)-pretreated mice[38,39], DBCO-insulin or unmodified insulin was intraperitoneally injected. At 8 h post the injection, a significantly higher level of insulin was detected in the plasma of mice injected with AAM and DBCO-insulin than the control mice (Fig. 7c, d), demonstrating the successful conjugation of DBCO-insulin to azido-labeled RBCs in vivo and subsequent release of insulin from RBCs. We have also monitored the blood concentration of insulin over an extended time. Mice were treated with AAM to enable in vivo RBC labeling, followed by i.p. injection of DBCO-ester-insulin. The blood concentration of insulin was measured via ELISA. For all the selected time points between 1 and 24 h post DBCO-ester-insulin injection, AAM-treated mice exhibited a higher plasma insulin level than PBS-treated mice (Supplementary Fig. 13f), supporting the prolonged blood circulation of drugs via our RBC labeling and targeting approach. However, it is noteworthy that the ester linkage between DBCO and insulin is easily degradable under in vivo conditions, and we envision the tunability of drug release kinetics with the use of different chemical linkages. In the subsequent glucose tolerance test, AAM-mediated RBC labeling combined with DBCO-insulin improved the control of blood glucose levels, in comparison with the non-targeting groups (Fig. 7e, f and Supplementary Fig. 13g). In consistence with the glucose control result, diabetic mice treated with AAM and DBCO-insulin showed the fastest recovery in body weight than other groups (Fig. 7g, h). These experiments demonstrated the promise of the RBC labeling and targeting technology to enhance the pharmacokinetics and therapeutic efficacy of drugs.

## Discussion

Metabolic glycan labeling has been widely explored for the engineering of cancer cells, immune cells, stem cells, and other types of

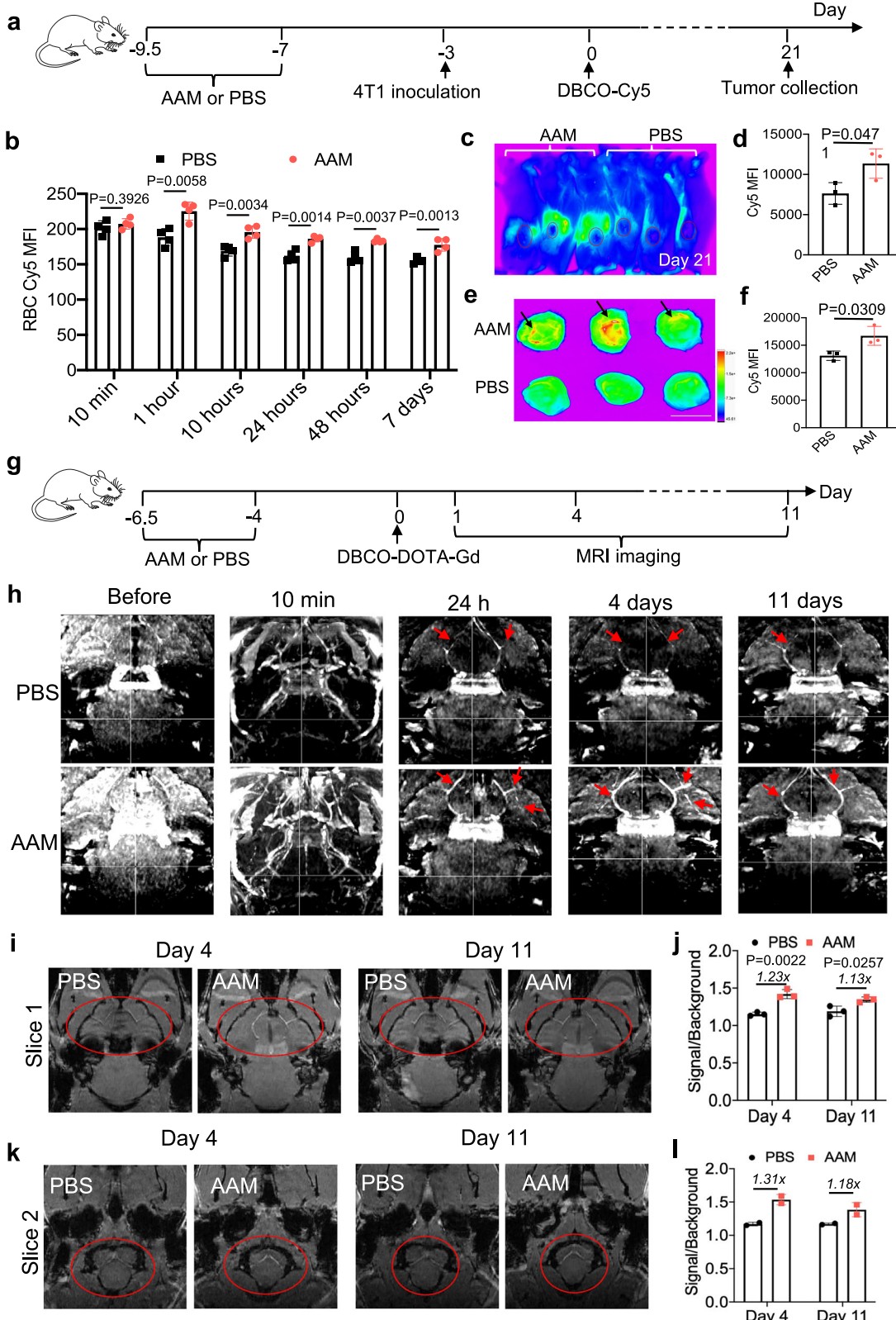

cells[29–37]. However, compared to these highly proliferative cells, nucleus-free RBCs undergo much lower metabolic activity and limited proliferation, casting a doubt on the feasibility of metabolic glycan labeling of RBCs. Nevertheless, glycoproteins and glycolipids are still integral components of RBC membranes and are synthesized in the cells via the metabolic glycoengineering pathways[40–42], which we hypothesized can be utilized to introduce chemical tags to RBC membranes. Despite showing a relatively lower labeling efficiency than other reported cell types, AAM managed to metabolically label RBCs with azido groups in vitro and in vivo. For in vivo labeling of circulating RBCs, AAM administration yields ~10–15% azide+ RBCs, with room for further optimization. AAM is not specific to RBCs and can also label WBCs with azido groups. However, the density of cell-surface azido groups on WBCs showed a rapid decay, and the number of azide+ RBCs

**Fig. 6 | In vivo conjugation of fluorophore and Gd agent to RBCs enables long-term fluorescence imaging and MRI of blood vessels and tissues. a–f** RBC labeling and targeting, whole blood imaging, and 4T1 tumor imaging. AAM or PBS ($n = 4$ mice per group) was i.v. injected twice daily for three days, followed by i.v. injection of DBCO-Cy5 on day 0 and subcutaneous injection of 4T1 tumor cells on day 17. **a** Study timeline. **b** Cy5 fluorescence intensity of RBCs at different times post injection of DBCO-Cy5 ($n = 4$ mice per group). **c** IVIS imaging of Balb/c mice at 21 days post injection of DBCO-Cy5. **d** Cy5 fluorescence intensity of tumors in (**c**) ($n = 3$ mice per group). **e** Ex vivo imaging of 4T1 tumors harvested at 21 days post injection of DBCO-Cy5. Scale bar: 1 cm. The blood vessel is indicated by the arrow. **f** Cy5 fluorescence intensity of 4T1 tumors in (**e**) ($n = 3$ mice per group). **g** Timeline for RBC labeling and targeting and MRI studies. AAM or PBS was i.v. injected twice daily for three days, followed by i.v. injection of DBCO-DOTA-Gd and MRI scans.

**h** Coronal view of 3D maximum intensity projection images of AAM- or PBS-treated mice at different times. Blood vessels are indicated by red arrows. **i, k** Coronal view comparison of AAM- and PBS-treated mice at 4 or 11 days post injection of DBCO-DOTA-Gd. The blood vessels of interest are indicated by red circles. **j, l** Contrast enhancement analysis of coronal view 1 **i** and view 2 (**k**), respectively. The signal from blood vessels and the background were quantified by ImageJ. Three visible vessels were analyzed for (**j**), and two visible vessels were analyzed for (**l**). All the numerical data are presented as mean ± SD. Multiple unpaired $t$-tests with Holm–Sidak correction for multiple comparisons were used for (**b, d, f, j**). ($0.01 < *P \leq 0.05$; $**P \leq 0.01$; $***P \leq 0.001$; $****P \leq 0.0001$). All experiments were performed at least twice independently with similar results. Source data for this figure is available in the Source Data file.

was 93-fold, 583-fold, and 3,844-fold of azide$^+$ WBCs at 2.5, 4.5, and 7.5 days, respectively, post AAM injections (Fig. 2h–k). At 4 days post AAM injections, cells in healthy tissues, including liver, kidney, brain, lung, heart, and spleen, also showed minimal azido labeling (Fig. 4i–k). These data reduce the concern of non-specific labeling and enable relatively specific targeting of RBCs as long as DBCO-cargos are administered 3 or more days after AAM injections. Future development of unnatural sugars that can preferentially metabolically label RBCs will further improve the labeling and targeting specificity of RBCs.

The in vivo safety profile of azido-sugars such as AAM is well supported by past studies[32–37]. In this study, we further demonstrated the minimal impact of AAM on the morphology, size, and metabolic states of RBCs (Fig. 4a–d) and populations of leukocytes in the bloodstream (Fig. 4e–g). Neither was any noticeable toxicity observed in tissues including liver, spleen, heart, lung, and kidney (Fig. 4h). The long presence of azido-labeled RBCs in the bloodstream (up to 42 days) also indicates the minimal immunogenicity of azido groups introduced onto RBCs. Compared to conventional RBC functionalization methods such as antibody conjugates and LPXTG chemistry[19,26–28], AAM-mediated RBC labeling method is easier, safer, more compatible with in vivo settings, and likely more efficient.

Due to its long life-span and excellent tissue accessibility, RBC has been an active engineering target to potentially improve the blood retention and tissue accumulation of diagnostic and therapeutic agents. In addition to the abovementioned antibody and LPXTG chemistries, nanoparticles or molecules that can non-specifically bind to circulating RBCs have also been actively explored[11,43,44], which are limited by the poor specificity and binding efficiency. In our approach, at 3 days post injection of AAM, DBCO-cargos can be specifically targeted to circulating RBCs via click chemistry (Fig. 5a–d). The conjugated cargos such as Cy5 can well retain on RBCs and circulate in the bloodstream for over 35 days (Fig. 5a–d). This is arguably the best RBC-hijacking strategy to date. We demonstrate that the conjugation of fluorophores to circulating RBCs can light up blood vessels and enable long-term fluorescence imaging of blood vessels and tissues, including tumors (Fig. 6a–f). In addition to fluorophores, MRI contrast agents such as Gd can also be conjugated to RBCs in vivo. Due to the prolonged blood retention of RBC-conjugating Gd, MRI could detect enhanced blood vessel signals for over 11 days with a single dose of Gd (Fig. 6g–l). In current practice, MRI contrast agents need to be administered at minutes to several hours prior to each MRI scan[45]. The ability to scan blood vessels for weeks after a single dose of contrast agents provides possibilities for deciphering the evolving vasculatures in both neonatal and mature tissues.

The RBC labeling and targeting technology also provides a facile and universal strategy to improve the pharmacokinetics of drugs, including small-molecule doxorubicin and macromolecular insulin (Fig. 7 and Supplementary Figs. 12 and 13). Drugs, upon systemic administration, are often cleared from the bloodstream within minutes to hours, posing a hurdle for the therapeutic efficacy and necessitating

frequent dosing of drugs. By metabolically tagging RBCs with azido groups and conjugating drugs to RBCs in vivo, drugs can circulate in the bloodstream for days to weeks and can be gradually released from RBCs with the incorporation of a cleavable linkage. We detected a higher level of insulin in the plasma of AAM-treated mice than control mice at 8 h post administration of DBCO-insulin (Fig. 7c, d). The enhanced pharmacokinetics of insulin led to improved control of blood glucose levels and alleviation of type-1 diabetes (Fig. 7e–h). We envision further enhancement in the pharmacokinetics and therapeutic efficacy of drugs, with the optimization of in vivo RBC labeling efficiency, in vivo DBCO-drug conjugation efficiency, and drug release kinetics from RBCs. In principle, any drug of interest, with simple DBCO functionalization, can be conjugated to azido-tagged RBCs in vivo to tune the overall pharmacokinetics. Several prior studies utilized i.v. injection of N-hydroxysuccinimide (NHS)-functionalized agents for covalent coupling to circulating RBCs[46], but the non-specific non-bioorthogonal NHS-amine chemistry presents a tremendous safety concern by non-specifically tagging all types of proteins, lipids, sugars, and nucleic acids.

While we have not tested metabolic glycan labeling of human RBCs, the well-known presence of glycoproteins and glycolipids in human RBCs adds to the feasibility. Indeed, human RBCs mainly express Neu5Ac (AAM labeling pathway) while murine RBCs express both Neu5Ac and Neu5Gc[47], suggesting that AAM-mediated metabolic glycan labeling could be more efficient in human RBCs. Given the much longer lifespan of human RBCs (~120 days), we also anticipate that azido-labeled RBCs could retain longer in the bloodstream than azido-labeled murine RBCs. Future studies on in vitro and in vivo labeling of human RBCs are needed to fully assess the safety and translational potential of the RBC labeling and targeting technology.

We envision the RBC tagging and targeting technology as promising for long-term imaging and certain drug delivery applications. Conventional contrast agents (either small molecules or nanomaterials) are short-acting and are often cleared from the circulation within hours. For scenarios when long-term monitoring is needed, e.g., continuous monitoring of disease progression, the necessity for repeated dosing of contrast agents adds to the overall cost, complexity, and the likelihood to lose track of the dynamics of vascularization in diseased sites. Our RBC labeling and targeting approach, instead, enables the circulation of contrast agents in the bloodstream and thus continued monitoring of vasculatures for over 4 weeks. Likewise, for drug delivery, endogenous azido-tagged RBCs could retain the clicked cargos for an extended time than conventional drug delivery vehicles such as nanoparticles and microparticles, potentially leading to a reduced dosing frequency of drugs and thus reduced side effects

Our in vivo RBC metabolic labeling approach offers a minimally invasive and modular strategy for RBC functionalization, but several limitations warrant attention. Differences between murine and human RBCs may impact the overall labeling efficiency and cargo retention time, and the potential immunogenicity upon repeated dosing of cargos remains to be fully assessed. Also, the amounts of cargos,

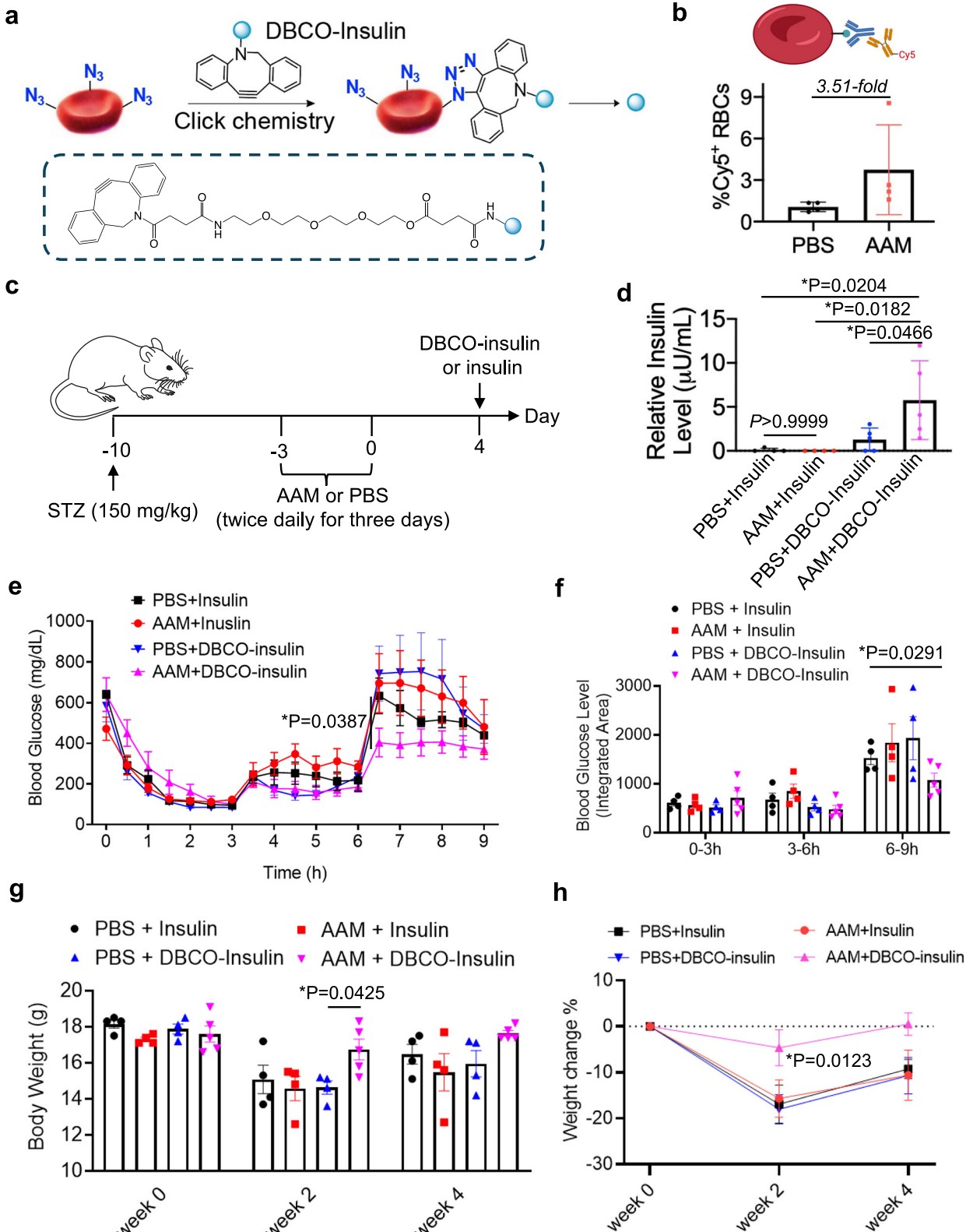

especially high-molecular-weight cargos, that can be conjugated to circulating RBCs are still limited. For example, the amount of DBCO-DOTA-Gd and DBCO-insulin that could be conjugated to circulating azido-labeled RBCs was relatively low, leading to a modest efficacy. The definite number of azido tags per RBC is certainly a main factor, but the physicochemical properties of DBCO-cargos and the chemical linkage between DBCO and cargo also require further optimization. Compared to existing RBC engineering methods, our approach

bypasses the process of isolating RBCs, engineering RBCs ex vivo, and infusing the engineered RBCs, but further optimization of the labeling and targeting specificity and efficiency is needed to realize the translational potential.

To summarize, we have developed an unprecedented in vivo RBC labeling and targeting technology that enables metabolic labeling of circulating RBCs with chemical tags (e.g., azido groups). The labeling of mature RBCs and RBC precursor cells in the bone marrow both

**Fig. 7 | In vivo conjugation of insulin to RBCs improves the pharmacokinetics and blood glucose control. a** Schematic illustration for conjugation of DBCO-insulin to azido-labeled RBCs and subsequent release of insulin. **b** ELISA analysis of azido groups on the surface of RBCs that were collected at 14 days post injections of AAM or PBS (*n* = 3 mice per group). RBCs were fixed to ELISA plates, incubated with DBCO-biotin, and further treated with streptavidin-HRP, followed by the addition of HRP substrate and absorption measurement at 450 nm. Illustration was created with BioRender. **c** Time frame of type-1 diabetes study in an STZ-induced diabetic mouse model. **d** Plasm insulin levels at 8 h post i.p. injection of DBCO-insulin. The insulin level was determined via a commercial ELISA kit (*n* = 4–5 mice per group).

**e**, **f** Glucose tolerance test (GTT) (*n* = 4–5 mice per group). Mice were fasted for 12 h, and then 10IU/kg DBCO-insulin or insulin was i.p. injected. Glucose was i.p. injected at 3 and 6 h. **f** Accumulated blood glucose levels between 0–3 h, 3–6 h, and 6–9 h (*n* = 4–5 mice per group). **g** Body weight of mice (*n* = 4–5 mice per group) at 0, 2, and 4 weeks post the injection of DBCO-insulin or insulin. **h** Percentage changes of mouse body weight over time for each group (*n* = 4–5 mice per group). All the numerical data are presented as mean ± SD. Two-tailed Welch's *t*-test was used for (**b**, **e**, **f**, **g**, **h**); one-way ANOVA with post hoc Fisher's LSD test was used for (**d**) (0.01 <*P* ≤ 0.05; **P* ≤ 0.01; ***P* ≤ 0.001; ****P* ≤ 0.0001). Source data for this figure is available in the Source data file.

contribute to the overall RBC labeling efficiency. Azido-tagged RBCs can circulate in the bloodstream for over 42 days in mice and enable the targeted conjugation of DBCO-cargos via efficient click chemistry. The conjugated cargos can retain on RBCs for over 35 days, and with the incorporation of a cleavable linkage between DBCO and cargo, can be gradually released into the bloodstream. This unprecedented RBC labeling and targeting technology provides a strategy to probe and understand RBC biology, prolong the blood circulation of contrast agents for long-term imaging of blood vessels and tissues, and improve the pharmacokinetics and therapeutic efficacy of drugs.

## Methods

### Ethical statement
This research complies with all relevant ethical regulations. All procedures involving animals were done in compliance with National Institutes of Health and Institutional guidelines with approval from the Institutional Animal Care and Use Committee at the University of Illinois at Urbana-Champaign (Protocol #24177).

### Materials and instrumentation
Mannosamine hydrochloride, galactosamine hydrochloride, glucosamine hydrochloride, dicyclohexyl carbodiimide, *N*-hydroxysuccinimide, and other chemical reagents were purchased from Sigma Aldrich (St. Louis, MO, USA) unless otherwise noted. Primary antibodies used in this study were obtained from Thermo Fisher Scientific (Waltham, MA, USA). All antibodies were diluted according to the manufacturer's recommendations. Fixable viability dye efluor780 was obtained from Thermo Fisher Scientific (Waltham, MA, USA). SYTO™ 85 Orange Fluorescent Nucleic Acid Stain was purchased from Thermo Fisher Scientific (Waltham, MA, USA). ENLITEN® ATP Assay System Bioluminescence Detection Kit was purchased from Thermo Fisher Scientific (Waltham, MA, USA). Histopaque®-1077 was purchased from Sigma Aldrich (St. Louis, MO, USA). Halt™ Protease Inhibitor Cocktail was purchased from Thermo Fisher Scientific (Waltham, MA, USA). Nitrocellulose membrane was purchased from Thermo Fisher Scientific (Waltham, MA, USA). Insulin ELISA kit (Catalog 80-INSHU-E01.1) was purchased from ALPCO (Salem, NH, USA). Clarity BG1000 Blood Glucose Meter and Strips were purchased from VWR International, LLC (Radnor, PA, USA). FACS analyses were collected on Attune NxT or BD LSR Fortessa flow cytometers and analyzed on FlowJo v7.6 and FCS Express v6 and v7. Fluorescent images were taken with a EVOS microscope (Thermofisher, Waltham, MA, USA). Confocal laser scanning microscopy (CLSM) images were taken by using a Zeiss LSM 700 Confocal Microscope (Carl Zeiss, Thornwood, NY, USA). High-performance liquid chromatography (HPLC) analysis was performed on a Shimadzu CBM-20A system (Shimadzu, Kyoto, Japan) equipped with an SPD-20A PDA detector (190–800 nm), an RF-20A fluorescence detector, and an analytical C18 column (Shimadzu, 3 μm, 50 × 4.6 mm, Kyoto, Japan). Nuclear Magnetic Resonance (NMR) spectra were recorded on a Varian U500 (500 MHz) or VXR500 (500 MHz), or a Bruker Carver B500 (500 MHz) spectrometer. Electrospray ionization (ESI) mass spectra were obtained from a Waters ZMD Quadrupole Instrument (Waters, Milford, MA, USA). Matrix-assisted laser desorption/ionization (MALDI) spectra were collected on

the Bruker Ultraflextreme MALDI-TOF/TOF Mass Spectrometer. In vivo and ex vivo images of animals and tissues were taken on a Bruker In Vivo Imaging System (Bruker, Billerica, MA, USA). Preparative HPLC was performed on a CombiFlash®Rf system (Teledyne ISCO, Lincoln, NE, USA) equipped with a RediSep®Rf HP C18 column (Teledyne ISCO, 30 g, Lincoln, NE, USA). Lyophilization was conducted in a Labconco FreeZone lyophilizer (Kansas City, MO, USA). Magnetic resonance imaging (MRI) scans were performed on a 9.4 T Bruker 30 cm AVANCE NEO equipped with a dual channel transmit, 4 receiver channels, a 4-channel mouse brain array, and a 4-channel rat brain array coil. The following fluorophore-conjugated anti-mouse antibodies were used: PE-anti-CD45 (clone 30-F11; Thermo Fisher Scientific, cat. no. 12-0451-82; dilution 1: 200), PE-Cy7-anti-CD44 (clone IM7; Thermo Fisher Scientific, cat. no. 25-0441-82; dilution 1:200), and FITC-anti-TER119 (clone TER-119; Thermo Fisher Scientific, cat. no. 11-5921-82; dilution 1:200). Phosphatidylserine staining was performed using Annexin V-FITC (Thermo Fisher Scientific, cat. no. A13199; dilution 1:200). All antibodies and reagents were validated by the manufacturer for flow cytometry applications.

### Cell lines and animals
The MEL cell line was a generous gift from Dr. Martin Burke's lab at the University of Illinois at Urbana-Champaign. The murine breast carcinoma cell line 4T1 (female, BALB/c origin; ATCC® CRL-2539™) and the murine melanoma cell line B16-F10 (male, C57BL/6 origin; ATCC® CRL-6475™) were obtained from the American Type Culture Collection (ATCC, Manassas, VA, USA). MEL cells were cultured in DMEM containing 10% FBS, 100 units/mL Penicillin G, non-essential amino acids, and 5% BSA at 37 °C in 5% $CO_2$ humidified air. 4T1 and B16F10 cells were cultured in DMEM containing 10% FBS, 100 units/mL Penicillin G at 37 °C in 5% $CO_2$ humidified air. Cell lines were used at low passage after thawing and were routinely tested for mycoplasma contamination using a commercial detection kit, and were confirmed negative prior to experiments. Female C57BL/6 and Balb/c mice (6–8 weeks) were purchased from the Jackson Laboratory (Bar Harbor, ME, USA). Only Female mice were used in this study, due to the challenge in housing a large number of male mice. Feed and water were available ad libitum. Artificial light was provided in a 12 h/12 h cycle.

### Synthesis of AAM, AAGal, and AAGlu
Mannosamine hydrochloride or galactosamine hydrochloride or glucosamine hydrochloride (1.0 e.q.) was dissolved in anhydrous methanol and cooled to 0 °C, followed by the addition of sodium methoxide in methanol (1.0 e.q.). Chloroacetic anhydride (1.05 e.q.) and triethylamine (1.0 e.q.) were then added. The reaction mixture was stirred at room temperature for overnight. Upon removal of the solvent, the crude product was re-dissolved in water. Sodium azide (4.0 e.q.) was then added, and the reaction mixture was stirred at 60 °C for overnight. After removing the solvent, the crude product was re-dissolved in pyridine, followed by the addition of acetic anhydride and 4-dimethylaminopyridine. After 24 h, the solvent was removed and the crude product was purified via silica column chromatography using ethyl acetate and hexane as the eluents. ¹H NMR of AAM (CDCl₃, 500 MHz): δ (ppm) 6.66&6.60 (d, *J* = 9.0 Hz, 1H, C(O)N*H*CH),

6.04&6.04 (d, 1H, $J = 1.9$ Hz, NHCHC*H*O), 5.32-5.35&5.04-5.07 (dd, $J = 10.2$, 4.2 Hz, 1H, CH₂CHC*H*CH), 5.22&5.16 (t, $J = 9.9$ Hz, 1H, CH₂CHCHC*H*), 4.60-4.63&4.71-4.74 (m, 1H, NHC*H*CHO), 4.10-4.27 (m, 2H, C*H*₂CHCHCH), 4.07 (m, 2H, C(O)C*H*₂N₃), 3.80-4.04 (m, 1H, CH₂C*H*CHCH), 2.00-2.18 (s, 12H, C*H*₃C(O)). ¹³C NMR (CDCl₃, 500 MHz): δ (ppm) 170.7, 170.4, 170.3, 169.8, 168.6, 168.3, 167.5, 166.9, 91.5, 90.5, 73.6, 71.7, 70.5, 69.1, 65.3, 65.1, 62.0, 61.9, 52.8, 52.6, 49.9, 49.5, 21.1, 21.0, 21.0, 20.9, 20.9, 20.9, 20.8. ESI MS (*m/z*): calculated for $C_{16}H_{22}N_4O_{10}Na$ [M+Na]⁺ 453.1, found 453.1.

## Mass spectrometry

Electrospray ionization mass spectrometry (ESI-MS) was performed to confirm the molecular mass of the synthesized chemical compounds. A total of one purified sample was analyzed ($n = 1$), as the analysis was conducted solely for compound identity verification. For sample preparation, the compound was dissolved in methanol or water based on the solubility, and 10 µL of the solution was directly injected into the mass spectrometer. The mobile phase consisted of 50% methanol containing 0.1% formic acid. Mass spectra were acquired using a Waters Quattro Ultima (Waters Corporation) operated in electrospray ionization positive mode (ESI⁺) with direct infusion. Raw data were acquired and processed using MassLynx software (Waters Corporation, v4.1) using default processing parameters, including automatic centroid detection, Savitzky–Golay smoothing, and automated peak picking. The resulting spectra were manually inspected, and the dominant molecular ion peak was annotated. The observed *m/z* value was compared with the calculated molecular mass of the target compound to validate compound identity.

## General procedure of flow cytometry

Cells were prepared according to individual experimental designs. Flow cytometry analyses were performed on an Attune NxT flow cytometer (Thermo Fisher Scientific) equipped with 405 nm, 488 nm, 561 nm, and 640 nm lasers. Instrument settings were optimized using unstained and single-stained controls and kept constant within each experiment. Spectral compensation was calculated using single-color controls and applied uniformly to all samples. Acquisition thresholds were set on forward scatter (FSC) to exclude debris. Debris and dead cells were excluded based on FSC/SSC parameters and viability dye staining, and doublets were removed using FSC-A versus FSC-H gating. Cy5 fluorescence was detected in the APC channel (640 nm excitation, 660/20 nm emission filter). A minimum of 10,000 live singlet events were collected per sample. Data were analyzed using FlowJo (BD Biosciences). Mean fluorescence intensity (MFI) or percentage of positive cells was quantified as indicated in the figure legends. All experiments were independently repeated at least three times with similar results.

## In vitro metabolic labeling of MEL cells

MEL cells were cultured in DMEM containing non-essential amino acids, 10% FBS, and 5% BSA at the density of $5 \times 10^5$/mL for 12 h. AAM or AAGal or AAGlu with a final concentration of 50 or 100 µM was added to the culture medium. At selected time points, cells were collected, washed with opti-MEM twice, and incubated with DBCO-Cy5 (10 µg/mL) and SYTO™ 85 Orange Fluorescent Nucleic Acid Stain (5 µM) at room temperature for 30 min. After washing, cells were analyzed on a flow cytometer. To initiate the differentiation of MEL cells, 2% DMSO and 50 µM iron citrate were added. The counts and size of cells were analyzed via a cell-counter, and hemoglobin was detected by analyzing UV absorption at 438 nm.

## In vitro metabolic labeling of primary RBCs

Blood was collected from C57BL/6 or Balb/c mice, and RBCs were harvested after the removal of lymphocytes. To maintain the viability of mouse RBCs in vitro, DMEM media containing non-essential amino acids, 10% FBS, 5% BSA, and 10 g/L glucose were used. RBCs were seeded at $10^8$/mL in 24-well plates in the presence of 50 or 100 µM AAM or AAGal. After 24 h, $10^7$ RBCs were taken out, washed with Alsever's solution twice, and incubated with DBCO-Cy5 (10 µg/ml) and SYTO™ 85 Orange Fluorescent Nucleic Acid Stain (5 µM) at room temperature for 1 h. Cells were washed and analyzed on a flow cytometer.

## In vivo metabolic labeling of RBCs

C57BL/6 or Balb/c mice were intravenously injected with AAM (100 mg/kg) or PBS twice daily for 3 consecutive days. At 1, 3, 5, 7, 14, 21, 28, 35, 42, and 49 days post the last injection of AAM or PBS, blood was collected and stored in Alsever's solution. After washing, cells were stained with DBCO-Cy5 (5 µg/mL) and SYTO™ 85 Orange Fluorescent Nucleic Acid Stain (5 µM) at room temperature for 1 h. After washing with Alsever's solution for three times, samples were run on a flow cytometer. For some experiments, AAM (200 mg/kg) was i.p. injected for 2, 4, and 6 times with an interval of 12 h. For the analysis of RBC precursor cells in the bone marrow, at 48 h post the last injection of AAM, bone marrow cells in the tibia and femur of mice were collected and stained with DBCO-Cy5, anti-Ter119, anti-CD44, and fixable viability dye, prior to flow cytometry analysis. To detect azido-labeled cells in healthy tissues, tissues were sectioned with a thickness of 20 µm, stained with DBCO-Cy5 and DAPI for 20 min, washed with PBS for multiple times, and mounted onto a microscope slide with the addition of Prolong Gold, prior to confocal imaging.

## Enrichment of azido-labeled RBCs

C57BL/6 mice were intravenously injected with AAM (100 mg/kg) or PBS twice daily for 3 consecutive days. At 7 days post the injection of AAM, RBCs were harvested, incubated with DBCO-desthiobiotin for 1 h, and then incubated with streptavidin-functionalized magnetic beads. Cells were then passed through a MACS cell separation column to remove unbounded RBCs (flow-through fraction). The magnetic bead-bounded RBCs were then eluted out using a solution of biotin.

## Fluorescence imaging of RBCs

Blood was drawn from C57BL/6 mice at selected times post the injection of AAM or PBS, and kept in the Alsever's solution. After washing, RBCs were incubated with 5 µg/mL DBCO-Cy5 at room temperature for 1 h, washed with Alsever's solution for 5 times, and added to a glass coverslip. Samples were stored at 4 °C prior to fluorescence imaging.

## RBC agglutination analysis

C57BL/6 mice were intravenously injected with AAM (100 mg/kg) or PBS twice daily for 3 days. At different times after the final AAM injection, blood was extracted from the tail of mice and diluted with Alsever's solution. A fraction of cells was directly imaged under a fluorescence microscope for morphology and agglutination analysis. Cells were also analyzed on a flow cytometer. The singlets were quantified from the FSC-A/FSC-H plot.

## ATP level measurement

C57BL/6 mice were intravenously injected with AAM (100 mg/kg) or PBS twice a day for 3 days. At different times after the final AAM injection, blood was extracted from the tail of mice and washed with Alsever's solution twice. The cell pellet was suspended in ATP-free water, followed by three freeze-thaw cycles to lyse RBCs. The ATP level in RBC lysates was measured using the ENLITEN® ATP Assay System Bioluminescence Detection Kit.

## Resorufin assay of RBCs

C57BL/6 mice were intravenously injected with AAM (100 mg/kg) or PBS twice a day for 3 days. At different times after the final AAM injection, blood was extracted from the tail of mice and washed with Alsever's solution twice. The cell pellet was resuspended in Alsever's

solution and incubated with resazurin (50 μg/mL) at 37 °C for 2 h, respectively. The resorufin product was detected on a plate reader ($\lambda$ex/$\lambda$em = 560/590 nm). RBCs incubated with $H_2O_2$ at room temperature for 10 min were used as the negative control.

## Osmotic fragility and mechanical fragility test

C57BL/6 mice were intravenously injected with AAM (100 mg kg$^{-1}$) or PBS twice daily for 3 consecutive days. Blood was collected 3 days after the final injection. For the AAM-treated group, red blood cells (RBCs) were separated into azido-positive and azido-negative populations using the above-described enrichment method for azido-labeled RBCs. These two fractions, together with RBCs from PBS-treated mice, were subjected to osmotic and mechanical fragility assays. For the osmotic fragility assay, $1 \times 10^6$ RBCs were pelleted by centrifugation and resuspended in 100 μL NaCl solutions of varying concentrations. After incubation at room temperature for 30 min, samples were centrifuged to pellet intact RBCs. Hemolysis was quantified by measuring the absorbance of the supernatant at 415 nm and normalized to complete lysates achieved by incubation of RBCs in ACK lysis buffer. For the mechanical fragility assay, $1 \times 10^7$ RBCs in 1 mL PBS from each group were transferred to tubes containing ten 4-mm glass beads and placed horizontally on an orbital shaker at 60 rpm. At the indicated time points, 100 μL aliquots were collected and centrifuged to remove intact RBCs. Hemolysis was determined by measuring the absorbance at 415 nm and expressed as a percentage of total lysis relative to ACK-treated samples.

## In vivo targeting of RBCs

C57BL/6 mice were intravenously injected with AAM (100 mg/kg) or PBS twice a day for 3 days. At 3 days post the last injection of AAM, DBCO-Cy5 (5 mg/kg) was intravenously injected. At selected times, blood was collected and placed in the Alsever's solution. After washing, cells were incubated with SYTO™ 85 Orange Fluorescent Nucleic Acid Stain (5 μM) at room temperature for 30 min, washed, and analyzed on a flow cytometer.

## Western blot analysis

C57BL/6 mice were intravenously injected with AAM (100 mg/kg) or PBS twice daily for 3 days. Blood was collected at 7 days post the final injection of AAM or PBS. White blood cells were removed with Histopaque®-1077 according to the manufacturer's protocol. RBCs were lysed with 0.1x PBS containing Halt™ Protease Inhibitor Cocktail at 4 °C for 30 min, followed by centrifugation at $21,000 \times g$ for 30 min and washing with 0.1x PBS containing Halt™ Protease Inhibitor Cocktail for 5 times to yield the RBC membranes. RBC membranes were lysed and incubated with alkyne-PEG$_4$-biotin for overnight at 4 °C. After washing with 0.1x PBS, proteins were loaded into SDS-PAGE gels for electrophoresis and subsequent transfer to the nitrocellulose membrane. The nitrocellulose membrane was blocked with the blocking buffer (5% non-fat milk in PBST) for overnight at 4 °C, incubated with streptavidin-HRP for 30 min, washed with PBST for 5 times, and treated with HRP substrate for chemiluminescence imaging on an ImageQuant 800. Protein bands were also stained with Ponceau S. For some analyses, RBCs or proteins of RBCs were already conjugated with DBCO-desthiobiotin. Gel electrophoresis followed by desthiobiotin detection will be directly performed for those samples.

## RBC lipid extraction and analysis

C57BL/6 mice were intravenously injected with AAM (100 mg/kg) or PBS twice daily for 3 days. Blood was collected at 7 days post the final injection of AAM or PBS. After removing the white blood cells with Histopaque®-1077, RBCs were incubated with DBCO-Cy5 (10 μg/mL) in Alsever's solution for 1 h at room temperature. Cells were then washed with Alsever's solution for three times and resuspended in a mixture of methanol and dichloromethane (3/7, v/v). The samples were then sonicated for 10 min. The protein precipitate was removed by centrifugation at $21,000 \times g$ for 10 min at 4 °C. The lipid-containing supernatant was gently collected and analyzed on an HPLC.

## Ex vivo retention of Cy5 on RBC surface

RBCs were isolated from C57BL/6 mice at 7 days post injections of AAM or PBS, and incubated with DBCO-Cy5 (5 μg/mL) for 1 h at room temperature. Cells were washed three times and resuspended in Alsever's solution. An aliquot of sample was directly analyzed on the flow cytometer, and the remaining sample was stored at 4 °C. At selected times, an aliquot of each sample was run on the flow cytometer.

## In vivo fluorescence imaging of blood vessels and tumors

For B16F10 tumor study, C57BL/6 mice were intravenously injected with AAM (100 mg/kg) or PBS twice a day for 3 days. DBCO-Cy5 (5 mg/kg) was intravenously injected at 4 days post the final injection of AAM or PBS. Blood was drawn from the tail at selected times for fluorescence imaging. At 17 days post DBCO-Cy5 injection, $5 \times 10^6$ B16F10 cells were subcutaneously injected into the flank of C57BL/6 mice. Tumors were collected 7 days after inoculation and subjected to IVIS imaging. For 4T1 tumor study, Balb/c mice were intravenously injected with AAM (100 mg/kg) or PBS twice daily for 3 days. At 4 days post the final injection of AAM or PBS, $2 \times 10^6$ 4T1 cells were subcutaneously injected into the flank of mice. After 3 days, DBCO-Cy5 (5 mg/kg) was i.p. injected. Blood was drawn at selected times for fluorescence imaging and flow cytometry analysis. IVIS imaging of mice was also performed at selected times. At 21 days post tumor cell inoculation, tumors and organs were harvested for IVIS imaging. During the 4T1 tumor imaging study, at selected times, blood was also collected for the analysis of Cy5 signal in the plasma. The blood cells were removed by centrifugation at $2000 \times g$ for 10 min, and the supernatant was harvested for Cy5 fluorescence measurement on a plate reader.

## Synthesis of DBCO-DOTA-Gd

1,4,7,10-Tetraazacyclododecane-1,4,7,10-tetraacetic acid (DOTA, 1.1 e.q.) and DBCO-PEG$_4$-amine (1.0 e.q.) were dissolved in dimethyl sulfoxide, followed by the addition of N,N-diisopropylethylamine (DIPEA) (4.4 e.q.) and hexafluorophosphate azabenzotriazole tetramethyl uronium (1.0 e.q.). After 2 h, the solvent was removed, and the residue was purified by preparative HPLC to yield DBCO-DOTA. DBCO-DOTA (1.0 e.q.) and DIPEA (3.5 e.q.) were then dissolved in water, followed by the addition of Gd(NO$_3$)$_3$·6H$_2$O (1.5 e.q.). After 4 h, the solvent was removed, and the residue was purified by preparative HPLC to yield DBCO-DOTA-Gd.

## MRI of mice

C57BL/6 mice were intravenously injected with AAM (100 mg/kg) or PBS twice a day for 3 days. At 4 days post the final injection of AAM or PBS, DBCO-DOTA-Gd (250 mg/kg) was intravenously injected. Mice were imaged before the injection and at different times after in a 9.4 T Bruker system (Bruker BioSpec 94/30 USR) using a 1H receive-only 2×2 mouse brain surface array coil. The mice were initially anesthetized with 2.5% isoflurane and maintained at 0.5–1.5% isoflurane throughout the experiment. Body temperature was held at 37 °C by circulating warm water and monitored using a rectal thermosensor. The Small Animal Monitoring & Gating System (SA Instruments) was utilized for continuous monitoring of both respiration rate and body temperature, ensuring the respiration rate was controlled within the range of 35 to 65 per minute. For contrast-enhanced magnetic resonance angiography (CE-MRA) to compare blood vessel visualization, a 3D T1-weighted FLASH sequence was used with coronal slices to minimize tight of flight effects. Scan parameters were: TE = 2.11 ms, TR = 15 ms, readout bandwidth = 59 kHz, number of averages = 4, flip angle = 15, FOV = $23.25 \times 14.23 \times 13.27$ mm$^3$, matrix size = $200 \times 180 \times 72$, and total scan time = 10 min. The same parameters were used for the scans.

## DBCO-doxorubicin synthesis

DBCO-NHS and DIPEA were dissolved in acetonitrile, followed by the addition of hydrazine monohydrate. The mixture was stirred for 30 min. After removing the solvent, the crude product was purified via preparative HPLC to yield DBCO-hydrazide. DBCO-hydrazide and doxorubicin were then dissolved in anhydrous methanol, followed by the addition of trifluoroacetic acid. After the complete consumption of DBCO-hydrazide, the solvent was removed under reduced pressure. The crude product was purified via preparative HPLC to yield DBCO-doxorubicin.

## DBCO-doxorubicin degradation

DBCO-doxorubicin was dissolved in buffers with a pH of 7.4, 6.5, and 5.5, respectively, and incubated at 37 °C. At selected times, 10 μL aliquots were run on an HPLC to determine the degradation kinetics of DBCO-doxorubicin.

## In vivo targeting of DBCO-doxorubicin to RBCs

C57BL/6 mice were intravenously injected with AAM (100 mg/kg) or PBS twice a day for 3 days. At 3 days post the final injection of AAM or PBS, DBCO-doxorubicin (5 mg/kg) in a mixture of DMSO/PBS (1/9, v/v) was i.v injected. At 2 or 24 h post the injection of DBCO-doxorubicin, blood was collected from mice and stored in Alsever's solution. After staining with SYTO™ 85 Orange Fluorescent Nucleic Acid Stain, cells were analyzed on a flow cytometer. Doxorubicin fluorescence signal in RBCs was detected via flow cytometry using the PE channel (488 nm excitation and 575 nm emission).

## DBCO-PE synthesis and conjugation to RBC

PE (1.0 e.q.) was dissolved in PBS, followed by the addition of DBCO-PEG$_4$-NHS (10 e.q.). The mixture was stirred at 4 °C overnight. After removing small molecules using the Sephadex G-25 resin column, the purified DBCO-PE was concentrated via ultracentrifugation with a 100 kDa ultra centrifugal filter. RBCs isolated from AAM- or PBS-treated mice were incubated with DBCO-PE in Alsever's solution for 1 h at room temperature. After washing with Alsever's solution three times, cells were stained with a DNA dye and analyzed on a flow cytometer.

## DBCO-OVA and DBCO-DNP-KLH synthesis and vaccination study

Ovalbumin (OVA) or dinitrophenyl–keyhole limpet hemocyanin (DNP-KLH) (1.0 equiv.) was dissolved in PBS, followed by the addition of DBCO-PEG4-NHS ester (10 equiv.). The reaction mixture was gently stirred at 4 °C overnight. Unreacted small molecules were removed using a Sephadex G-25 desalting column. Purified DBCO-OVA was concentrated by ultrafiltration using a 30 kDa molecular weight cutoff centrifugal filter, while DBCO-DNP-KLH was concentrated using a 100 kDa molecular weight cutoff centrifugal filter. For the in vivo humoral response study, C57BL/6 mice were intravenously injected with AAM (100 mg/kg) or PBS twice daily for 3 consecutive days. Four days after the final AAM or PBS administration, mice were intravenously injected with DBCO-OVA or DBCO-DNP-KLH. Blood samples were collected at the indicated time points, and plasma was isolated by centrifugation at $2000 \times g$ for 10 min to remove blood cells and platelets. Plasma samples were stored at −80 °C until analysis. After the final time point, all plasma samples were subjected to antibody titer analysis.

## ELISA for antigen-specific IgG1 and IgG2a antibody titers

Unless otherwise specified, all procedures were performed at room temperature. OVA or DBCO-DNP-KLH was diluted in PBS to a concentration of 10 μg/mL and used to coat high-binding 96-well ELISA plates overnight at 4 °C. Plates were washed three times with PBS and blocked with 2% (w/v) BSA in PBST (0.1% Tween-20 in PBS) for 1 h. After one wash with PBST, serially diluted plasma samples were added to the plates and incubated overnight at 4 °C. Plates were then washed three times with PBST and incubated with biotin-conjugated anti-mouse IgG1 or IgG2a antibodies diluted in blocking buffer for 2 h. Following three additional washes with PBST, streptavidin-HRP diluted in blocking buffer was added and incubated for 30 min. After washing the plates three times with PBST, the HRP substrate was added and allowed to develop for 15 min before the reaction was terminated with 2 M sulfuric acid. Absorbance was measured at 450 nm. Antibody titers were defined as the lowest plasma dilution yielding a signal above baseline, or alternatively, OD$_{450}$ values were directly used for quantitative comparison.

## Synthesis of DBCO-insulin

DBCO-ester-NHS was first synthesized. DBCO-PEG$_3$-OH (1.0 e.q.) and succinic anhydride (2.0 e.q.) were dissolved in anhydrous pyridine, followed by the addition of 4-dimethylaminopyridine (1.0 e.q.). The mixture was stirred for overnight. After removing the solvent, the crude product was purified using preparative HPLC to yield DBCO-ester-COOH. DBCO-ester-COOH (1.0 e.q.) was then dissolved in dichloromethane, followed by the addition of 1-ethyl-3-(3-dimethylaminopropyl)carbodiimide (1.2 e.q.) and N-hydroxysuccinimide (2.0 e.q.). The mixture was stirred overnight. After removing the solvent, the crude product was purified via silica gel column chromatography to yield DBCO-ester-NHS. The as-synthesized DBCO-ester-NHS (2.0 e.q.) and insulin (1.0 e.q.) were dissolved in PBS and stirred overnight at 4 °C. The crude product was purified via ultracentrifugation using a 3 kDa ultracentrifugal filter.

## In vitro conjugation of DBCO-insulin to cells

$5 \times 10^4$ 4T1 cells in 96-well plates were incubated with AAM (50 μM) or PBS for 24 h. Cells were detached with trypsin and incubated with DBCO-insulin (0.1 mg/mL) in FACS buffer (2% BSA in PBS) for 1 h at 4 °C. After washing, cells were stained with rabbit anti-insulin primary antibody for 30 min at 4 °C, and then incubated with Cy5-conjugated goat anti-rabbit secondary antibody for 30 min, prior to flow cytometry analysis. Similarly, RBCs isolated from AAM- or PBS-treated mice were incubated with DBCO-insulin in Alsever's solution (0.1 mg/mL) for 1 h, followed by staining with rabbit anti-insulin primary antibody and Cy5-conjugated goat anti-rabbit secondary antibody. Cells were also stained with a DNA dye. After washing, RBCs were analyzed on a flow cytometer.

## STZ induced type 1 diabetes (T1D)

To induce T1D, female C57BL/6 mice were injected with STZ (150 mg/kg). After monitoring the blood glucose levels for 7 days and confirming the onset of T1D (blood glucose level >220 mg/dL without fasting), mice were randomly divided into 4 groups. Mice were intravenously injected with AAM (100 mg/kg) or PBS twice daily for 3 days. Four days later, DBCO-insulin or insulin (10 IU/kg) was i.p injected. For the measurement of plasma insulin levels, blood was collected at 8 h post injection of DBCO-insulin or insulin and placed in heparin-coated tubes. After removing blood cells via centrifugation, the supernatant was harvested for the quantification of insulin using the insulin ELISA kit (Catalog 80-INSHU-E01.1).

## Glucose tolerance test (GTT)

Female C57BL/6 mice were fasted for 12 h, and DBCO-insulin or insulin (10 IU/kg) was i.p injected (counted as time 0). At selected times, blood was drawn from the tail, and the glucose level in the blood was measured using the Clarity BG1000 Blood Glucose Meter and Strips. At 3 h and 6 h, 1.5 g/kg glucose was i.p injected to each mouse. The blood glucose level was continuously monitored for 9 h with an interval of 0.5 h. After the GTT experiment, DBCO-insulin or insulin was injected every week for 2 more injections (3 injections in total). The body weight of mice was closely monitored.

## Statistics and reproducibility

Flow cytometry data were analyzed using FlowJo (BD Biosciences, v10.8.1). Statistical analyses and data visualization were performed using GraphPad Prism (GraphPad Software, v9.5.1). For in vitro studies, each experiment included at least three biological replicates, and technical replicates were included where appropriate. For in vivo experiments, the number of animals (*n*) per group is specified in the corresponding figure legends. Animals were randomly assigned to experimental groups when applicable. Representative images (e.g., confocal microscopy, fluorescence imaging, histology) are shown from one experiment, and similar results were observed in at least three independent experiments or independent biological replicates. Statistical analyses were performed as described in the corresponding figure legends. Data are presented as mean ± SD. Exact *p* values and statistical tests are indicated in the figures or figure legends.

## Reporting summary

Further information on research design is available in the Nature Portfolio Reporting Summary linked to this article.

## Data availability

All data provided in this study can be found in the main text, figures, supplementary information, and source data files. Source data are provided with this paper.

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

## Acknowledgements

We acknowledge the support from National Science Foundation CAREER Award DMR 21-43673 CAR (H.W.), National Institutes of Health R01CA274738 (H.W.), National Institutes of Health R21CA270872 (H.W.), Sontag Distinguished Scientist Award (H.W.), American Cancer Society Award (H.W.), and the start-up package from the Department of Materials Science and Engineering at the University of Illinois at Urbana-Champaign and the Cancer Center at Illinois (H.W.). Y.L. acknowledges the support from the Cancer Center at Illinois Graduate Scholarship.

## Author contributions

Conceptualization: H.W. and Yusheng L. Methodology: H.W., Yusheng L., Yizun W., M.B., Yuan L. Formal analysis: Yuan L., K.K., H.H., and C.L. Investigation: Yusheng L., Yueji W., J.Z., D.B., J.H., R.B., D.N., C.L., M.B., and F.L. Visualization: Yusheng L., Yizun W., F.L., and H.W. Funding acquisition: H.W. Project administration: H.W. Supervision: H.W. and F.L. Writing—original draft: H.W., Yusheng L. Writing—review and editing: Yusheng L., Yizun W., K.K., Yuan L., Yueji W., H.H., J.Z., D.B., J.H., R.B., D.N., C.L., M.B., F.L., and H.W.

## Competing interests

H.W. and Y.L. have filed a patent application (No. 63/402,413) for the red blood cell metabolic labeling and targeting technology. Other authors declare no competing interests.
