## [Peer Review File · Nature Communications]

In Vivo Metabolic Tagging and Targeting of Circulating Red Blood Cells

Corresponding Author: Professor Hua Wang

Version 1:

Reviewer comments:

Reviewer #1

(Remarks to the Author)

Hua Wang and Co-workers in this paper have investigate the development and possible use of RBC for coupling different kind of agents of possibly diagnostic or therapeutic interest. The idea is not original since many labs have already produced and used RBC coupled to agents of interest. A number of reviews are available. Hua Wang et All. have investigate a possible glycan labelling producing data obtained in vitro and in vivo on murine RBC. Metabolic glycan labeling has been widely explored for the engineering of cancer cells, immune cells, stem cells, and other types of cells as highlighted in the reference list correctly reported in the manuscript. This approach is of potential general interest and a number of applications are shown.

The major limitation of the manuscript is the use only of mice RBC without any evidence that similar results could be translated to human RBC. This is not a trivial difference that should at least be discussed.

The second major concern involve the possible induced immunogenicity of the generated constructs. During the years it has been shown that aptens or peptides/proteins coupled to the RBC membrane can become immunogenic after one or more administrations in vivo. In some cases the modality of coupling and the nature of the agent coupled can instead induce immunotolerance.

To overcome these limitations potentially therapeutic agents have been coupled to RBC membranes by a number of different approaches (i.e. <https://doi.org/10.3390/pharmaceutics12050440>) but also encapsulated into the RBC to prevent drug inactivation by eventually generate antibodies (<https://doi.org/10.1080/17425247.2020.1822320>). These modalities should be mentioned since relevant to the scope of the paper.

A further question is related to the possibility of activate the RBC membrane in vivo. Apparently this is a new approach but others have previously administered in vivo agents that modify the RBC membrane with success (Blood, Vol 93, No 1 (January 1), 1999: pp 376-384).

The Discussion section should consider not only the potential benefits but also the limits of the approach (some examples are briefly mentioned above) and spend some time to provide to the reader a more complete view of the approaches developed by others in the field of RBC use as carriers for therapeutic or diagnostic agents.

It is not fear to write "Existing RBC engineering approaches are limited to isolated RBCs under ex vivo conditions, and the whole process of isolating RBCs, modifying RBCs ex vivo, and infusing engineered RBCs is time-consuming and costly, often causes damage to RBCs, and suffers from a high risk of infections.(Ref. 9,17-19)". A recent paper published il Lancet Neurology (DOI: 10.1016/S1474-4422(24)00220-5) reports data obtained in human patients with a rare disease highlighting safety and efficacy of a RBC-based drug delivery system in a large phase 3 clinical trial. Furthermore, references 17-19 are not appropriate referring to transfusion products not to engineered RBC.

Minor points:

Extended Fig. 3a and 3b: please explain the meaning of Cy5Fluorescence intensity axes in Fig.3a per μL blood. Is this figure showing the n. of Cy5+ cells or fluorescence intensity?

With respect to the percentage of RBC-AAM (and WBC-AAM) can you explain figures close to about 100% (Ext Fig 3b) when the main text explain that in vivo at 24h from AAM administration labelled RBC are 4.2%? If the Fig.3b represent the total number of RBC (and WBC) it is not clear at 7.5 days if in circulation we still have all labelled RBC or if 100% RBC is represented by new RBC entering in circulation to replace RBC-AAM that are removed from circulation as shown in Ext. Fig. 3a at day 7.5. Please comment.

Extended data Fig.4: please explain Fig.4b. % refer to labelled cells or total cells? If the figure refer to total cells this is not representative of the successful labelling of RBC precursors but simply of RBC precursors distribution number upon AAM

labelling versus PBS. Please clarify

Intraperitoneal injections: the dose administered to increase the % of labelled RBC to max 14% is really very high totaling about 1.2g/Kg of AAM administered! Please comment.

In vivo toxicity: The determination of RBC main properties obtained from cells of animals receiving iv AAM are of limited interest since only a small percentage of said RBC are modified by AAM under the experimental conditions used. In other words measurements of ATP or NAD/NADH are representative of all RBC present in the samples and not of the labelled RBC. This should be commented. In addition, Fig. 4j clearly shows that heart and kidney are significantly labelled by AAM. In vivo targeting of DBCO cargo: please comment Extended Data Figure 6. The differences among day 0 and day 1; in addition explain while day 5 Cy5 FI ratio appears to decrease respect day 3.

Long-term fluorescence imaging: labelled RBC have been administered i.p. and not i.v., please comment. Is this modality more appropriate for targeting/labelling implanted tumor cells?

MRI imaging of brain blood vessels: Assuming that Gd coupled to RBC is stable were Gd ends when disappear from circulation? Is this removed in spleen and liver? Does cause any kidney toxicity? Please address these comments.

In vivo conjugation of drugs to RBCs: DBCO-doxorubicin intravenously administered can label only about 1% of RBC at 2 hours and is not significant versus the administration of PBS. These results are of very limited interest and doesn't prove any potential therapeutic benefit. The same apply to coupling of PE. In this case DBCO-functionalized PE is added ex-vivo for two hours to AAM activated RBC. The same apply to DBCO-insulin detected with rabbit anti insulin and Cy5-conjugated goat anti-rabbit secondary antibody on RBC after an ex-vivo incubation of two hours. The difference versus mice receiving PBS is not significantly different. In vivo experiments with peritoneal administration of DBCO-insulin in AAM treated STR mice show a limited effect only at one time point questioning the benefit of this claimed therapeutic treatment. This should be commented in more details highlighting the limitations of the approach.

Reviewer #2

(Remarks to the Author)

The study by Yusheng Liu et al describe a novel approach for RBC's engineering with the potential application in imaging and drug delivery.

Overall, the stud is well designed and executed, following are few comments that could further benefit the work:

1. RBCs are critical for gas exchange, endothelial functions, platelet reactivity, thrombosis and haemorrhage, binding to pathogens, scavenging chemokines. Furthermore, the elasticity of RBC's is essential for vascular functions specially at the presence of prooxidants s. These functions require through testing to ensure the competence of RBCs after azido tagging and cargo incorporation. Tests such as membrane integrity tests (osmotic and Mechanical), oxidative haemolysis assay, Ektacytometry, and RBC's indices such as MCV, RDW, MCH and MCHC; can augment the safety profile of the proposed strategy.
2. The main proposed usages mentioned in the studies are, imaging and drug delivery. For imaging, the reasoning for having a long-term tagging needs to be justified compared to classical imaging techniques which relies on short acting contrast agents. Similarly for RBCs as a drug carrier, more discussion is needed to justify the advantages of this pathway compared to using exogenous carriers (e.g. nanoparticles).
3. The study demonstrated improved pharmacokinetics for drugs like insulin, however, the release kinetics of conjugated drugs from RBCs are not thoroughly characterized. This is critical for ensuring controlled and predictable therapeutic outcomes.

Reviewer #3

(Remarks to the Author)

This manuscript from the Wang lab describes the labeling of red blood cells (RBCs) with azido sugars for various potential downstream applications. The authors convincingly show that RBCs can be metabolically labelled with ManNAz (termed AAM here) both in vitro and in vivo. Additionally, the subsequent bioorthogonal reaction (SPAAC) can be performed in vivo as well. All of this confirms data from the Bertozzi lab and others concerning these sugars and SPAAC reaction. While the levels of the labeling are modest, the interesting and new feature here is that the azide-sugar persists on the cell surface of RBCs much longer than on other cells and tissues, allowing for a level of selectivity based on the timing between AAM and DBCO delivery. Using this feature the authors go on to show that they can detect differences in vascularization by fluorescence and image blood vessels by MRI. Finally, they conjugate pharmaceutical cargos to the RBCs and nicely show that insulin can act longer when appended to RBCs. Again, the magnitude of the differences here are mostly small, but I believe this is a potentially interesting application of metabolic engineering that is worth further exploration, particularly in terms of cargo circulation and delivery. I am supportive of publication. I do have two comments that should be addressed first.

Comment #1 - I suggest that the authors are more explicit about achieving RBC selectivity by waiting longer between AAM delivery and subsequent treatment with DBCO. I understand that they talk about the unique retention of azides on RBCs versus other tissues several times in the paper. However, I think this could be made more clear for non-experts, so that readers do not come away with the impression that other tissues are not labeled at all.

Comment #2 - How did the authors detect DOX on cells in Figure 9d? Was this antibody-based as well. I did not see the details for this experiment.

Reviewer #4

(Remarks to the Author)

The submitted study reports *in vivo* metabolic glycan labeling of red blood cells (RBCs) using systemically administered azido-sugars. The authors claim that RBCs are tagged with azides as cellular surface glycans (glycoproteins and glycolipids). The authors showed the surface azido tags on RBCs persist >42 days, whereas tags on leukocytes and other cells rapidly decay. The labeled RBCs can capture DBCO-linked cargos by click chemistry, prolonging a cargo's circulation from hours to >35 days. As examples, one dose of azido-sugar plus a DBCO-gadolinium agent yields months-long brain-vasculature MRI, and linking insulin to tagged RBCs extends its blood half-life. In summary, the paper claims a stable *in vivo* RBC-labeling platform that dramatically extends cargo persistence and enables new imaging and drug-delivery applications.

While the study demonstrates solid technical execution and offers long-lived labeling of circulating RBCs using azido sugars and click chemistry, it fundamentally builds upon well established strategies already reported by the same research group. The core methodology, systemic delivery of peracetylated azido-monosaccharides for metabolic labeling and subsequent conjugation of DBCO-bearing molecules via SPAAC, has been applied previously to dendritic cells, platelets, and adipocytes. In each case, similar labeling and *in vivo* targeting were made. The difference in this RBC study from previous is not the chemical mechanism, but merely the target cell type (RBCs) and the extended persistence of the azido label. While the longer half-life of labeled RBCs is notable (~42 days vs ~4 days for platelets), this reflects more a known biological property of RBCs than an engineered advance. Thus, the central novelty lies in demonstrating expected behavior in a new context. Moreover, the applications presented, such as prolonged imaging or drug delivery, are anticipated outcomes of this platform, not surprising or mechanistically new.

Although the study convincingly demonstrates that azido groups persist on the surface of circulating RBCs for over 42 days, it does not adequately establish that this labeling strategy is selective enough for practical applications in targeted imaging or drug delivery. The authors argue that azide tags decay rapidly in leukocytes and other nucleated blood cells within 3 days, thereby enabling selective RBC targeting using DBCO-functionalized molecules if administered after this window. However, this argument overlooks a more critical issue: whether normal tissues and vasculature are also being labeled and whether that labeling persists to a degree that compromises selectivity. In particular, Figure 5i-j shows detectable fluorescence signals in normal tissues (liver, spleen, etc.) after DBCO-Cy5 injection, even in the AAM-treated group. Although the signal is lower than in RBCs, it is not negligible, and there is no quantification relative to background or functional implications of this residual labeling. The authors use a "wait 3 days post AAM" strategy to reduce leukocyte labeling, but it remains unclear whether this delay also fully eliminates labeling in non-blood tissues. The Western blot and confocal data in Fig. 4i-j are helpful but limited, they do not resolve whether low-level but functionally significant tagging persists. Furthermore, the authors' own prior studies on metabolic labeling of platelets, adipocytes, and dendritic cells reported relatively broad azido labeling, including some tissue types. The current claim that RBCs uniquely retain azido tags while all other tissues rapidly clear them seems inconsistent with those earlier findings, suggesting either differing experimental standards or unexplored biological variability. This inconsistency undermines the claim of RBC-specific labeling and suggests that more rigorous, quantitative tissue-distribution studies are needed.

Overall, The study is technically well-conducted and shows promise for translational applications, but it does not introduce a sufficiently novel concept or mechanism to meet the publication standards of Nature Communications. The core advances are too directly on a sequence of prior works from the same group using near-identical strategies. There is no demonstration of fundamentally new biological insight, molecular mechanism, or chemical methodology that would justify publication in a journal prioritizing broad conceptual advances. In addition, the study does not sufficiently demonstrate RBC-specific labeling or targeting, and several results raise the possibility of non-negligible off-target effects in normal tissues. This undermines the claimed value of the system for selective imaging or drug delivery. Additionally, the conclusions appear partially inconsistent with earlier work from the same group on metabolic labeling of other cell types. Without stronger and more systematic evidence of selectivity, this work does not meet the robustness or clarity needed for publication in Nature Communications.

Version 2:

Reviewer comments:

Reviewer #1

(Remarks to the Author)

During the revision process the Authors have addressed many critical points and provided new data and explaining some of the limits of this approach. This is appreciable, although not conclusive, and support acceptance from the point of view of this Reviewer.

Main section: The Authors have amended the previous version of the paper that now read "...Significant progress has indeed been made to address these issues and bring RBC-based drug delivery systems to clinical trials.^{20,21}". In addition "While direct encapsulation of cargos into RBCs has been extensively explored, most approaches rely on *ex vivo* engineering of RBCs,^{48,49} which inevitably inherits the common issues in RBC stability, infections, costs during the isolation, storage, *ex vivo* manufacturing, and infusion processes." I respectfully disagree. The technology nowadays has been greatly improved and was not only taken into phase 3 clinical trial but the trial has also documented the safety of this approach in hundreds of patients (see Ref 21). This sentence should be amended.

Ref 20 is not correct since reports the title of Ref. 21

Immunogenicity:

New sentence "We are aware of the concern on the potential immunogenicity of cargos being conjugated to circulating RBCs. In separate studies, we attempted to conjugate DBCO-functionalized 2,4-dinitrophenylated Hemocyanin, Keyhole Limpet (DNP-KLH) and DBCO-ovalbumin (OVA), two representative antigens, to RBCs in vivo and monitor the antibody titers over time. As a result, the in vivo conjugation of DNP-KLH and OVA antigens failed to exert any enhancement on the antibody responses compared to the control groups (Fig. S9), which lessens the concern of potent immunogenicity of RBC-conjugated cargos in our metabolic labeling approach."

The new experiment is done by administering 0.1 mg of DNP-OVA or DNP-KLP that likely will partially be bound to activated RBC while partially will remain free in circulation. I'm encouraging the Authors to provide an estimate of the percentage of the administered antigen that is RBC conjugated vs. the total administered.

Reviewer #3

(Remarks to the Author)

The authors have addressed my minor comments appropriately. I am supportive of publication.

Reviewer #4

(Remarks to the Author)

The authors have thoroughly and convincingly addressed all major and minor concerns raised during the review process. In particular, the revised manuscript now includes substantially strengthened discussion and new experimental data addressing the translational relevance to human RBCs, immunogenicity concerns, RBC membrane integrity and hematological safety, and limitations and positioning of the approach relative to existing RBC engineering strategies.

The additional supplementary figures enhance the rigor and credibility of the work. Importantly, the authors now provide a balanced and transparent discussion of both the strengths and limitations of the in vivo metabolic RBC labeling platform, which greatly improves clarity for the reader and appropriately contextualizes the novelty of the approach.

While the underlying chemistry is established, the demonstration of long-term, in vivo metabolic labeling of circulating RBCs and its exploitation for extended imaging and drug circulation represents a meaningful and technically challenging advance with clear implications for RBC-based diagnostics and therapeutic delivery. The manuscript is now well aligned with the scope of the journal and meets the expected standards of technical quality, interpretation, and presentation.

I therefore recommend acceptance of the manuscript in its current form, without further revision.

Point-by-point reply to Reviewers' comments

(All responses were colored in blue; all changes in the manuscript were highlighted in yellow)

Review #1

Hua Wang and Co-workers in this paper have investigated the development and possible use of RBC for coupling different kind of agents of possibly diagnostic or therapeutic interest. The idea is not original since many labs have already produced and used RBC coupled to agents of interest. A number of reviews are available. Hua Wang et al. have investigated a possible glycan labelling producing data obtained *in vitro* and *in vivo* on murine RBC. Metabolic glycan labeling has been widely explored for the engineering of cancer cells, immune cells, stem cells, and other types of cells as highlighted in the reference list correctly reported in the manuscript. This approach is of potential general interest and a number of applications are shown.

Response: We thank the Reviewer for the positive comments, and addressed the remaining comments in the revised manuscript.

1. The major limitation of the manuscript is the use only of mice RBC without any evidence that similar results could be translated to human RBC. This is not a trivial difference that should at least be discussed.

Response: We appreciate this comment from the Reviewer. We have attempted to test the metabolic labeling of human RBCs, but have been experiencing trouble (mostly on IRB side) in obtaining fresh human RBCs. For now, we may not be able to fully address this comment experimentally, although we are fairly confident that the metabolic glycan labeling should work for human RBCs. We have now added some discussion on the applicability towards human RBCs in the revised manuscript: “While we have not tested metabolic glycan labeling of human RBCs, the well-known presence of glycoproteins and glycolipids in human RBCs adds to the feasibility. Indeed, human RBCs mainly express Neu5Ac (AAM labeling pathway) while murine RBCs express both Neu5Ac and Neu5Gc,⁵¹ suggesting that AAM-mediated metabolic glycan labeling could be more efficient in human RBCs. Given the much longer lifespan of human RBCs (~120 days), we also anticipate that azido-labeled RBCs could retain longer in the bloodstream than azido-labeled murine RBCs. Future studies on *in vitro* and *in vivo* labeling of human RBCs are needed to fully assess the safety and translational potential of the RBC labeling and targeting technology.” [Ref. 51: Varki, A. Loss of N-glycolylneuraminic acid in humans: mechanisms, consequences, and implications for disease. *Proc. Natl Acad. Sci. USA* 98, 5379–5386 (2001)]

2. The second major concern involve the possible induced immunogenicity of the generated constructs. During the years it has been shown that aptens or peptides/proteins coupled to the RBC membrane can become immunogenic after one or more administrations *in vivo*. In some cases the modality of coupling and the nature of the agent coupled can instead induce immunotolerance.

Response: We thank the Reviewer for this comment. In the current manuscript, we mainly reported small-molecule DBCO-functionalized agents (fluorophores, gadolinium, insulin) which are generally lowly immunogenic. Indeed, we attempted to utilize the RBC labeling and targeting technology for vaccine applications, with a hope that the longer retention of antigens in the bloodstream may prolong the antibody response. We tried to conjugate DBCO-DNP-

KLH and DBCO-OVA, two representative antigens, to circulating RBCs and monitored the antibody titers in the blood over time. As shown in the new Fig. S9, none of the conjugation groups showed enhanced antibody responses in comparison with the control groups.

We have now added Fig. S9 and the following text in the revised manuscript: “We are aware of the concern on the potential immunogenicity of cargos being conjugated to circulating RBCs. In separate studies, we attempted to conjugate DBCO-functionalized 2,4-dinitrophenylated Hemocyanin, Keyhole Limpet (DNP-KLH) and DBCO-ovalbumin (OVA), two representative antigens, to RBCs in vivo and monitor the antibody titers over time. As a result, the in vivo conjugation of DNP-KLH and OVA antigens failed to exert any enhancement on the antibody responses compared to the control groups (Fig. S9), which lessens the concern of potent immunogenicity of RBC-conjugated cargos in our metabolic labeling approach.”

Figure S9. In vivo targeting of antigens to circulating RBCs does not enhance the antibody responses. (a) Timeframe of DNP-KLH vaccination study. C57BL/6 mice were i.v. injected with AAM or PBS twice daily for three consecutive days. After labeling, mice were i.v. injected

with DBCO-DNP-KLH, and serum antibody titers were monitored over time. (b) Serum anti-DNP-KLH IgG1 antibody titers at indicated time points. (c) Serum anti-DNP-KLH IgG2a levels as determined by ELISA. (d) Timeframe of ovalbumin (OVA) vaccination study. C57BL/6 mice were i.v. injected with AAM or PBS twice daily for three consecutive days. After labeling, mice were i.v. injected with DBCO-OVA, and serum antibody titers were monitored over time. Shown are anti-OVA (e) IgG1 and (f) IgG2a levels at indicated time points. ELISA data are presented as OD₄₅₀ values at the indicated serum dilution. All the numerical data are presented as mean ± SD.

3. To overcome these limitations potentially therapeutic agents have been coupled to RBC membranes by a number of different approaches (i.e. <https://doi.org/10.3390/pharmaceutics12050440>) but also encapsulated into the RBC to prevent drug inactivation by eventually generate antibodies (<https://doi.org/10.1080/17425247.2020.1822320>). These modalities should be mentioned since relevant to the scope of the paper.

Response: We appreciate this comment from the Reviewer. We have now expanded the discussion on prior RBC engineering methods and cited the relevant references: “Non-covalent methods involve physical adsorption of cargos onto RBC membranes via electrostatic interactions^{11,22,23} or binding of a targeting ligand to RBC receptors (e.g., Ter119).^{13,24-26} The former is limited by the weak interactions between cargos and RBCs and the need of custom-designed adsorbable structures,^{11,22,23} while the latter is limited by the lack of specificity and low abundance of RBC receptors.^{13,24-26}” “While direct encapsulation of cargos into RBCs has been extensively explored, most approaches rely on ex vivo engineering of RBCs,^{48,49} which inevitably inherits the common issues in RBC stability, infections, costs during the isolation, storage, ex vivo manufacturing, and infusion processes. Several prior studies utilized i.v. injection of N-hydroxysuccinimide (NHS)-functionalized agents for covalent coupling to circulating RBCs,⁵⁰ but the non-specific non-bioorthogonal NHS-amine chemistry presents a tremendous safety concern by non-specifically tagging all types of proteins, lipids, sugars, and nucleic acids”.

48 Glassman, P. M., Villa, C. H., Ukidve, A., Zhao, Z., Smith, P., Mitragotri, S., Russell, A. J., Brenner, J. S. & Muzykantov, V. R. Vascular drug delivery using carrier red blood cells: focus on RBC surface loading and pharmacokinetics. *Pharmaceutics* 12, 440 (2020).

49 Rossi, L., Pierigè, F., Bregalda, A. & Magnani, M. Preclinical developments of enzyme-loaded red blood cells. *Expert Opin. Drug Deliv.* 18, 43–54 (2021).

50 Rettig, M. P., Low, P. S., Gimm, J. A., Mohandas, N., Wang, J. & Christian, J. A. Evaluation of biochemical changes during in vivo erythrocyte senescence in the dog. *Blood* 93, 376–384 (1999).

4. A further question is related to the possibility of activate the RBC membrane in vivo. Apparently this is a new approach but others have previously administered in vivo agents that modify the RBC membrane with success (*Blood*, Vol 93, No 1 (January 1), 1999: pp 376-384).

Response: We thank the Reviewer for this comment. Previous work by Rettig et al. (*Blood*, 1999) investigated the biochemical changes occurring during RBC senescence in dogs. They used NHS-biotin to modify circulating RBCs with biotin using the NHS-amine chemistry, with a goal of retrieving the RBCs for ex vivo analysis. While their goal was to tag and isolate RBCs, their data indicated that the biotinylation of RBCs did not notably alter the cellular function. As our bioorthogonal click chemistry approach is more selective than NHS-amine chemistry,

we anticipate that our approach will also be safe and do not alter RBC functions, as supported by our data in this manuscript.

Also, in response to another Reviewer's comment regarding potential perturbation of membrane integrity or function due to our labeling strategy, we have now included additional experimental data demonstrating that the metabolic tagging of RBCs with azido groups does not alter RBC membrane properties such as mechanical and osmotic fragility (**Fig. S6**).

Figure S6. In vivo metabolic glycan labeling of RBCs does not affect the osmotic and mechanical integrity of RBCs. C57BL/6 mice were i.v. injected with AAM or PBS twice daily for three consecutive days. Four days after the final injection, peripheral blood was collected and RBCs were isolated for functional assessment. (a) Osmotic fragility assay showing percentage of hemolysis as a function of NaCl concentration. (b) Mechanical fragility assay measuring hemolysis following rotational stress over time. Data are presented as mean \pm SD. Hemolysis was calculated as the ratio of supernatant absorbance at 415 nm (hemoglobin) to the absorbance at 415 nm obtained from an equal number of RBCs fully lysed with the ACK lysis buffer.

5. The Discussion section should consider not only the potential benefits but also the limits of the approach (some examples are briefly mentioned above) and spend some time to provide to the reader a more complete view of the approaches developed by others in the field of RBC use as carriers for therapeutic or diagnostic agents.

Response: We thank the Reviewer for this comment. We have now revised the Discussion accordingly. Specifically, we have now included discussions on the applicability of our labeling and targeting technology to human RBCs, the possibility of immune responses arising from repeated administrations of cargos, and limitations related to the diversity and size of cargos that can be efficiently conjugated and delivered using our platform: “While we have not tested metabolic glycan labeling of human RBCs, the well-known presence of glycoproteins and glycolipids in human RBCs adds to the feasibility. Indeed, human RBCs mainly express Neu5Ac (AAM labeling pathway) while murine RBCs express both Neu5Ac and Neu5Gc,⁵¹ suggesting that AAM-mediated metabolic glycan labeling could be more selective and efficient in human RBCs. Given the much longer lifespan of human RBCs (~120 days), we also anticipate that the azido-labeled RBCs could retain longer in the bloodstream than azido-labeled murine RBCs. Future studies on in vitro and in vivo labeling of human RBCs are needed to fully assess the safety and translational potential of the RBC labeling and targeting technology.” “Our in vivo RBC metabolic labeling approach offers a minimally invasive and modular strategy for RBC functionalization, but several limitations warrant attention. Differences between murine and human RBCs may impact the overall labeling efficiency and

cargo retention time, and the potential immunogenicity upon repeated dosing of cargos remains to be fully assessed. Also, the amounts of cargos, especially high-molecular-weight cargos, that can be conjugated to circulating RBCs could still be limited due to the definite number of azido tags per RBC. Compared to existing RBC engineering methods, our approach bypasses the process of isolating RBCs, engineering RBCs ex vivo, and infusing the engineered RBCs, but further optimization of the labeling specificity and efficiency is needed to realize the translational potential.”

6. It is not fear to write “Existing RBC engineering approaches are limited to isolated RBCs under ex vivo conditions, and the whole process of isolating RBCs, modifying RBCs ex vivo, and infusing engineered RBCs is time-consuming and costly, often causes damage to RBCs, and suffers from a high risk of infections.(Ref. 9,17-19)”. A recent paper published in *Lancet Neurology* (DOI: 10.1016/S1474-4422(24)00220-5) reports data obtained in human patients with a rare disease highlighting safety and efficacy of a RBC-based drug delivery system in a large phase 3 clinical trial. Furthermore, references 17-19 are not appropriate referring to transfusion products not to engineered RBC.

Response: We appreciate this comment from the Reviewer. We have now updated the Introduction: “Existing RBC engineering approaches mostly rely on isolating and modulating RBCs under ex vivo conditions, and the whole process of isolating RBCs, modifying RBCs ex vivo, and infusing engineered RBCs is time-consuming and costly, could cause damage to RBCs, and suffers from a risk of infections.^{9,17-19} Significant progress has indeed been made to address these issues and bring RBC-based drug delivery systems to clinical trials.^{20,21} In parallel, we envision direct in vivo engineering of RBCs will provide new possibilities for understanding RBC biology and developing enhanced diagnostics and therapies for a variety of diseases” We have also updated the references accordingly.

- 17 Klein, H. G., Spahn, D. R. & Carson, J. L. Red blood cell transfusion in clinical practice. *The Lancet* **370**, 415-426 (2007).
- 18 Magnani, M. & Rossi, L. Approaches to erythrocyte-mediated drug delivery. *Expert Opin. Drug Deliv.* **11**, 677–687 (2014).
- 19 Rossi, L. *et al.* Red blood cell membrane processing for biomedical applications. *Front. Physiol.* **10**, 1070 (2019).
- 20 Chiarantini, L. *et al.* Safety and efficacy of intra-erythrocyte dexamethasone sodium phosphate in children with ataxia telangiectasia (ATTeST): a multicentre, randomised, double-blind, placebo-controlled phase 3 trial. *Lancet Neurol.* **23**, 871–882 (2024).
- 21 Zielen, S. *et al.* Safety and efficacy of intra-erythrocyte dexamethasone sodium phosphate in children with ataxia telangiectasia (ATTeST): a multicentre, randomised, double-blind, placebo-controlled phase 3 trial. *Lancet Neurol.* **23**, 871–882 (2024).

Minor points:

7. Fig. S3a and S3b: please explain the meaning of Cy5 Fluorescence intensity axes in Fig.S3a per μL blood. Is this figure showing the n. of Cy5+ cells or fluorescence intensity?

Response: Fig. S3a presents the total Cy5 fluorescence intensity per microliter of blood, as calculated by ‘# of RBCs (or WBCs) per μL \times mean Cy5 fluorescence intensity of RBCs (or WBCs)’. Fig. S3b shows the percentage of RBCs or WBCs among all the blood cells at different time points. We have updated the figure captions to minimize confusion: “(a) Total

Cy5 fluorescence intensity of RBCs and WBCs that were harvested at different times and stained with DBCO-Cy5, as calculated by # of RBCs (or WBCs) per $\mu\text{L} \times$ mean Cy5 fluorescence intensity of RBCs (or WBCs). (b) Percentages of RBCs and WBCs among blood cells at different times post injections of AAM or PBS.”

8. With respect to the percentage of RBC-AAM (and WBC-AAM) can you explain figures close to about 100% (Fig. S3b) when the main text explain that in vivo at 24h from AAM administration labelled RBC are 4.2%? If the Fig. S3b represent the total number of RBC (and WBC) it is not clear at 7.5 days if in circulation we still have all labelled RBC or if 100% RBC is represented by new RBC entering in circulation to replace RBC-AAM that are removed from circulation as shown in Fig. S3a at day 7.5. Please comment.

Response: Fig. S3b shows the percentage of RBCs or WBCs among all the blood cells. The purpose of presenting Fig. S3b is to show that AAM treatment did not alter the blood cell fractions (i.e., not toxic). Figures in the main text refer to the percentages of azido-labeled RBCs among all the RBCs, showing the metabolic labeling efficiency of RBCs.

9. Fig. S4: please explain Fig. S4b. % refer to labelled cells or total cells? If the figure refer to total cells this is not representative of the successful labelling of RBC precursors but simply of RBC precursors distribution number upon AAM labelling versus PBS. Please clarify

Response: Similarly, Fig. S4b refers to total cells. The purpose is to show that AAM treatment did not change the populations of RBC precursors (i.e., not toxic). To avoid confusion, we have updated the Y axis of Fig. S4b.

Fig. S4b. Percentages of different RBC precursor cells among total erythroid cells at 2 days post injections of AAM or PBS.

10. Intraperitoneal injections: the dose administered to increase the % of labelled RBC to max 14% is really very high totaling about 1.2g/Kg of AAM administered! Please comment.

Response: For i.v. injections, 100 mg/kg per dose (6 doses in total) led to ~10% azido-labeled RBCs. For i.p., a higher dose could be used so we tested 200 mg/kg per dose. We believe the Reviewer is concerned about the safety of a seemingly high dose of AAM. Indeed, a dosage of 50-200 mg/kg AAM or similar azido-sugars with a total number of 3-9 doses has been commonly used in previous studies by other groups and our own group.

11. In vivo toxicity: The determination of RBC main properties obtained from cells of animals receiving iv AAM are of limited interest since only a small percentage of said RBC are

modified by AAM under the experimental conditions used. In other words measurements of ATP or NAD/NADH are representative of all RBC present in the samples and not of the labelled RBC. This should be commented.

Response: We appreciate the Reviewer’s comment. We did receive a lot of questions on the safety of our labeling technology, so the overall safety data will be helpful to our readers. Also, although only a fraction (~10%) of circulating RBCs is labeled under our current conditions, this proportion is still substantial enough that bulk measurements (ATP, NAD/NADH) would likely capture any major systemic or off-target effects of AAM treatment on overall RBC physiology. We agree with the Reviewer that characterizing the metabolic state of only the labeled RBCs would provide more specific insight. However, the process of purifying azido-labeled RBCs itself would alter RBC metabolism and introduce artifacts. Instead, we have additionally assessed RBC indices (MCV, MCH, MCHC, RDW), which revealed negligible differences between AAM and control groups. If AAM had notable effects on RBC physiology, changes in these indices would be expected even with ~10% labeled cells.

We have added Fig. S7 and the following text in the revised manuscript: “RBC indices including RBC counts, Hemoglobin concentration, hematocrit percentages, mean cell volume (MCV), mean corpuscular hemoglobin (MCH), mean corpuscular hemoglobin concentration (MCHC), and red cell distribution width (RDW) (Fig. S7a-g), as well as RBC morphology (Fig. S7h), showed negligible differences between AAM and PBS groups. These data demonstrated that AAM-mediated metabolic labeling induced negligible changes in RBC membrane fragility and RBC functions.”

Figure S7. In vivo metabolic glycan labeling of RBCs does not alter hematological indices or morphology of RBCs. C57BL/6 mice were i.v. injected with AAM or PBS twice daily for three consecutive days. Four days after the final injection, peripheral blood was collected for hematological analysis. Quantified RBC parameters include (a) RBC count, (b) hemoglobin concentration, (c) hematocrit, (d) mean corpuscular volume (MCV), (e) mean corpuscular hemoglobin (MCH), (f) mean corpuscular hemoglobin concentration (MCHC), and (g) red cell distribution width (RDW). (h) Representative RBC morphology assessment for individual

samples. All numerical data are presented as mean \pm SD.

12. In addition, Fig. 4j clearly shows that heart and kidney are significantly labelled by AAM.

Response: We would like to clarify that there was always some Cy5 background signal after we stained the tissue sections with DBCO-Cy5. Therefore, the difference between AAM and PBS groups, instead of the overall Cy5 signal, indicates the azido labeling efficiency. We also analyzed the average Cy5 fluorescence intensity; negligible differences between AAM and PBS groups were observed (Fig. 4k).

We have added Fig. 4k and the following text in the revised manuscript: “Confocal imaging of tissue sections, after staining with DBCO-Cy5, also showed negligible differences in Cy5 fluorescence intensity between AAM and PBS groups (Fig. 4j-k).”

Fig. 4k. Quantified Cy5 fluorescence intensity of tissues in Fig. 4j.

13. In vivo targeting of DBCO cargo: please comment Figure S6. The differences among day 0 and day 1; in addition explain while day 5 Cy5 FI ratio appears to decrease respect day 3.

Response: The difference between day 0 and day 1 could be attributed to the release of non-specifically bounded DBCO-Cy5. The Cy5 FI ratio changes between day 5 and day 3 could be due to the death of RBCs over a longer culture time.

14. Long-term fluorescence imaging: labelled RBC have been administered i.p. and not i.v., please comment. Is this modality more appropriate for targeting/labelling implanted tumor cells?

Response: To clarify, we i.p. injected DBCO-Cy5 (instead of labeled RBCs) into mice for the fluorescence imaging study. i.p. injection was chosen as it enables the sustained release of administered cargo over an extended time.

15. MRI imaging of brain blood vessels: Assuming that Gd coupled to RBC is stable were Gd ends when disappear from circulation? Is this removed in spleen and liver? Does cause any kidney toxicity? Please address these comments.

Response: It is true that DOTA-Gd could gradually dissociate and release Gd²⁺ over time. Free

Gd²⁺ is primarily cleared through the renal clearance, while intact Gd-conjugated RBCs (when become aged) are expected to be sequestered and cleared by the spleen and liver. In our study, we have not observed any sign of toxicity in major organs including kidneys. Also, mice showed no significant changes in body weight and no sign of adverse effects for over a year, supporting the benign safety profile for the dosage of DBCO-Gd that we used in the study.

16. In vivo conjugation of drugs to RBCs: DBCO-doxorubicin intravenously administered can label only about 1% of RBC at 2 hours and is not significant versus the administration of PBS. These results are of very limited interest and doesn't prove any potential therapeutic benefit.

Response: We appreciate the Reviewer's comment. We agree that the targeting effect of DBCO-Dox was not impressive. Nevertheless, we would like to present the representative data we have obtained to showcase the broad applicability of our RBC labeling and targeting technology, while also telling our readers there is significant room for improvement. Regarding the limited targeting efficiency of DBCO-Dox, the rapid clearance of injected DBCO-Dox and the potential premature release of conjugated Dox from RBCs could be among the reasons. Future studies should optimize the chemical linkages between DBCO and drugs and the overall physicochemical properties of DBCO-drugs.

17. The same apply to coupling of PE. In this case DBCO-functionalized PE is added ex-vivo for two hours to AAM activated RBC. The same apply to DBCO-insulin detected with rabbit anti insulin and Cy5-conjugated goat anti-rabbit secondary antibody on RBC after an ex-vivo incubation of two hours. The difference versus mice receiving PBS is not significantly different. In vivo experiments with peritoneal administration of DBCO-insulin in AAM treated STR mice show a limited effect only at one time point questioning the benefit of this claimed therapeutic treatment. This should be commented in more details highlighting the limitations of the approach.

Response: We thank the Reviewer for this comment. We fully agree with the Reviewer regarding the limited targeting efficiency of DBCO-cargos. These limitations are exactly what we aim to address in our ongoing and future work, in order to realize the full potential of our in vivo RBC labeling technology. We have now added more discussions on these limitations in the revised manuscript: "Our in vivo RBC metabolic labeling approach offers a minimally invasive and modular strategy for RBC functionalization, but several limitations warrant attention. Differences between murine and human RBCs may impact the overall labeling efficiency and cargo retention time, and the potential immunogenicity upon repeated dosing of cargos remains to be fully assessed. Also, the amounts of cargos, especially high-molecular-weight cargos, that can be conjugated to circulating RBCs are still limited. For example, the amount of DBCO-DOTA-Gd and DBCO-insulin that could be conjugated to circulating azido-labeled RBCs was relatively low, leading to a modest efficacy. The definite number of azido tags per RBC is certainly a main factor, but the physicochemical properties of DBCO-cargos and the chemical linkage between DBCO and cargo also require further optimization. Compared to existing RBC engineering methods, our approach bypasses the process of isolating RBCs, engineering RBCs ex vivo, and infusing the engineered RBCs, but further optimization of the labeling and targeting specificity and efficiency is needed to realize the translational potential."

The study by Yusheng Liu et al describes a novel approach for RBC's engineering with the potential application in imaging and drug delivery. Overall, the study is well designed and executed, following are a few comments that could further benefit the work:

Response: We thank the Reviewer for the positive comments, and addressed the remaining issues in the revised manuscript.

1. RBCs are critical for gas exchange, endothelial functions, platelet reactivity, thrombosis and haemorrhage, binding to pathogens, scavenging chemokines. Furthermore, the elasticity of RBC's is essential for vascular functions specially at the presence of prooxidants. These functions require through testing to ensure the competence of RBCs after azido tagging and cargo incorporation. Tests such as membrane integrity tests (osmotic and Mechanical), oxidative haemolysis assay, Ektacytometry, and RBC's indices such as MCV, RDW, MCH and MCHC; can augment the safety profile of the proposed strategy.

Response: We thank the Reviewer for this valuable comment. We have now supplemented the osmotic fragility and mechanical hemolysis analysis of azido-labeled RBCs vs control RBCs, by incubating RBCs in varied concentrations of sodium chloride or with varied mechanical rotation times. As shown in Fig. S6a-b, negligible differences between azido-labeled RBCs and control RBCs (azide-negative RBCs after AAM treatment or untreated RBCs) were observed. We have also analyzed key RBC indices including MCV, MCH, MCHC, and RDW, which showed negligible differences between AAM-treated RBCs and PBS-treated RBCs (Fig. S7). These results further support that azido tagging does not impair RBC membrane function or homeostasis under the conditions tested. Although we were unable to perform ektacytometry due to the lack of available equipment on campus, we have added a discussion of this limitation and the importance of comprehensive functional assessment for future translational efforts.

We have now added Fig. S6 and Fig. S7 and the following text in the revised manuscript: "We also analyzed the membrane osmotic and mechanical fragility of RBCs collected from AAM-treated mice (both azide-positive and azide-negative fractions) and PBS-treated mice, which showed negligible differences (Fig. S6a-b). RBC indices including RBC counts, Hemoglobin concentration, hematocrit percentages, mean cell volume (MCV), mean corpuscular hemoglobin (MCH), mean corpuscular hemoglobin concentration (MCHC), and red cell distribution width (RDW) (Fig. S7a-g), as well as RBC morphology (Fig. S7h), showed negligible differences between AAM and PBS groups. These data demonstrated that AAM-mediated metabolic labeling induced negligible changes in RBC membrane fragility and RBC functions."

Figure S6. In vivo metabolic glycan labeling of RBCs does not affect the osmotic and mechanical integrity of RBCs. C57BL/6 mice were i.v. injected with AAM or PBS twice daily for three consecutive days. Four days after the final injection, peripheral blood was collected and RBCs were isolated for functional assessment. (a) Osmotic fragility assay showing percentage of hemolysis as a function of NaCl concentration. (b) Mechanical fragility assay measuring hemolysis following rotational stress over time. Data are presented as mean \pm SD. Hemolysis was calculated as the ratio of supernatant absorbance at 415 nm (hemoglobin) to the absorbance at 415 nm obtained from an equal number of RBCs fully lysed with the ACK lysis buffer.

h

Morphology Feature	PBS-1	PBS-2	PBS-3	PBS-4	AAM-1	AAM-2	AAM-3	AAM-4
Polychromasia	1+	1+	1+	1+	1+	1+	1+	1+
Target Cells	Rare	-	-	-	-	-	-	-
Echinocytes	-	Rare	-	-	-	-	-	-
Hemoglobin Crystals	-	Rare	-	-	-	-	-	-
Howell-Jolly Bodies	-	-	-	-	Rare	Rare	Rare	Rare
Anisocytosis	-	-	-	-	Slight	Slight	Slight	-

Figure S7. In vivo metabolic glycan labeling of RBCs does not alter hematological indices or morphology of RBCs. C57BL/6 mice were i.v. injected with AAM or PBS twice daily for three consecutive days. Four days after the final injection, peripheral blood was collected for hematological analysis. Quantified RBC parameters include (a) RBC count, (b) hemoglobin concentration, (c) hematocrit, (d) mean corpuscular volume (MCV), (e) mean corpuscular hemoglobin (MCH), (f) mean corpuscular hemoglobin concentration (MCHC), and (g) red cell distribution width (RDW). (h) Representative RBC morphology assessment for individual samples. All numerical data are presented as mean \pm SD.

2. The main proposed usages mentioned in the studies are, imaging and drug delivery. For imaging, the reasoning for having a long-term tagging needs to be justified compared to classical imaging techniques which relies on short acting contrast agents. Similarly for RBCs as a drug carrier, more discussion is needed to justify the advantages of this pathway compared to using exogenous carriers (e.g. nanoparticles).

Response: We appreciate this comment from the Reviewer. We have now added more discussion in the revised manuscript to justify the promise of long-term RBC tagging: “We envision the RBC tagging and targeting technology is promising for long-term imaging and

certain drug delivery applications. Conventional contrast agents (either small molecules or nanomaterials) are short-acting and are often cleared from the circulation within hours. For scenarios when long-term monitoring is needed, e.g., continuous monitoring of disease progression, the necessity for repeated dosing of contrast agents adds to the overall cost, complexity, and the likelihood to lose track of the dynamics of vascularization in diseased sites. Our RBC labeling and targeting approach, instead, enables the circulation of contrast agents in the bloodstream and thus continued monitoring of vasculatures for over 4 weeks. Likewise, for drug delivery, endogenous azido-tagged RBCs could retain the clicked cargos for an extended time than conventional drug delivery vehicles such as nanoparticles and microparticles, potentially leading to a reduced dosing frequency of drugs and thus reduced side effects.”

3. The study demonstrated improved pharmacokinetics for drugs like insulin, however, the release kinetics of conjugated drugs from RBCs are not thoroughly characterized. This is critical for ensuring controlled and predictable therapeutic outcomes.

Response: We thank the Reviewer for this comment. We have now supplemented an in vivo pharmacokinetic study. Mice were treated with AAM to enable in vivo RBC labeling, followed by intraperitoneal injection of DBCO-ester-insulin. The blood concentration of insulin was monitored by ELISA. As shown in Fig. S13f, AAM-treated mice exhibited higher plasma insulin levels than PBS-treated mice at all selected time points between 1-24 h post injection of DBCO-ester-insulin, supporting the prolonged blood circulation of drugs via our RBC labeling and targeting approach. However, it is noteworthy that the ester linkage between DBCO and insulin is easily degradable under in vivo conditions, and we envision the tunability of drug release kinetics with the use of different chemical linkages.

We have now added Fig. S13f and the following text in the revised manuscript: “We have also monitored the blood concentration of insulin over an extended time. Mice were treated with AAM to enable in vivo RBC labeling, followed by i.p. injection of DBCO-ester-insulin. The blood concentration of insulin was measured via ELISA. For all the selected time points between 1 and 24 h post DBCO-ester-insulin injection, AAM-treated mice exhibited a higher plasma insulin level than PBS-treated mice (Fig. S13f), supporting the prolonged blood circulation of drugs via our RBC labeling and targeting approach. However, it is noteworthy that the ester linkage between DBCO and insulin is easily degradable under in vivo conditions, and we envision the tunability of drug release kinetics with the use of different chemical linkages.”

Fig. S13f. Blood concentration of insulin at different times post i.p. injection of DBCO-insulin

(10 IU/kg) in AAM-treated or PBS-treated mice.

Review #3

This manuscript from the Wang lab describes the labeling of red blood cells (RBCs) with azido sugars for various potential downstream applications. The authors convincingly show that RBCs can be metabolically labelled with ManNAz (termed AAM here) both in vitro and in vivo. Additionally, the subsequent bioorthogonal reaction (SPAAC) can be performed in vivo as well. All of this confirms data from the Bertozzi lab and others concerning these sugars and SPAAC reaction. While the levels of the labeling are modest, the interesting and new feature here is that the azide-sugar persists on the cell surface of RBCs much longer than on other cells and tissues, allowing for a level of selectivity based on the timing between AAM and DBCO delivery. Using this feature the authors go on to show that they can detect differences in vascularization by fluorescence and image blood vessels by MRI. Finally, they conjugate pharmaceutical cargos to the RBCs and nicely show that insulin can act longer when appended to RBCs. Again, the magnitude of the differences here are mostly small, but I believe this is a potentially interesting application of metabolic engineering that is worth further exploration, particularly in terms of cargo circulation and delivery. I am supportive of publication. I do have two comments that should be addressed first.

Response: We thank the Reviewer for the positive comments, we have addressed the remaining comments in the revised manuscript.

Comment #1 - I suggest that the authors are more explicit about achieving RBC selectivity by waiting longer between AAM delivery and subsequent treatment with DBCO. I understand that they talk about the unique retention of azides on RBCs versus other tissues several times in the paper. However, I think this could be made more clear for non-experts, so that readers do not come away with the impression that other tissues are not labeled at all.

Response: We thank the Reviewer for this considerate advice. We have added now clearly described the retention-based selectivity for our RBC labeling approach in the Introduction: "It is noteworthy that non-specific azido-sugars used in this study can label both RBCs and non-RBCs in vivo. However, due to the non-proliferative property and long life-span of RBCs, the installed azido groups are stable in the bloodstream for over 42 days, nearly the life-span (~45 days) of mouse RBCs. In contrast, the azido tags on the membrane of leukocytes in the blood and cells in healthy tissues such as liver, spleen, lung, heart, brain, and kidney decay to baseline levels within 3 days."

Comment #2 - How did the authors detect DOX on cells in Extended Data Figure 9d? Was this antibody-based as well. I did not see the details for this experiment.

Response: DOX was detected directly by flow cytometry using its intrinsic fluorescent property (488 nm excitation, 575 nm emission, PE channel). We have added this information to the Methods section as follows: "Doxorubicin fluorescence signal in RBCs was detected via flow cytometry using the PE channel (488 nm excitation and 575 nm emission)".

Review #4

The submitted study reports in vivo metabolic glycan labeling of red blood cells (RBCs) using systemically administered azido-sugars. The authors claim that RBCs are tagged with azides as cellular surface glycans (glycoproteins and glycolipids). The authors showed the surface azido tags on RBCs persist >42 days, whereas tags on leukocytes and other cells rapidly decay. The labeled RBCs can capture DBCO-linked cargos by click chemistry, prolonging a cargo's circulation from hours to >35 days. As examples, one dose of azido-sugar plus a DBCO-gadolinium agent yields months-long brain-vasculature MRI, and linking insulin to tagged RBCs extends its blood half-life. In summary, the paper claims a stable in vivo RBC-labeling platform that dramatically extends cargo persistence and enables new imaging and drug-delivery applications.

Response: We thank the Reviewer for the nice summary, and have attempted to address the remaining comments. Regarding the overall innovation, as an intentional shift from our prior work on rational design of trigger-activatable sugar derivatives for cell-selective labeling (which we have to admit is rather difficult and our designed sugar precursors are still far from satisfactory) [Nat Chem Biol 2017, 13, 415-424; Chem Commun 2018, 54, 4878-4881; PNAS 2023, 120, e2302342120], this work explores the possibility of leveraging the intrinsic properties of RBCs to achieve RBC-selective labeling in vivo. We envision the ability to metabolically tag and target circulating RBCs in vivo, as demonstrated in this study, will benefit the RBC research field and the development of RBC-based diagnostics and therapeutics. Also, while it seems a 'lazy' approach to apply metabolic glycan labeling to different types of cells in our lab, we have indeed focused on cells that are historically difficult to engineer in our recent efforts. Nucleus-free RBC without normal protein synthetic machinery is one such type of cells that metabolic glycan labeling was deemed unlikely; our work will clear those doubts and encourage researchers to explore metabolic labeling and targeting of many other types of cells that are difficult to engineer.

1. While the study demonstrates solid technical execution and offers long-lived labeling of circulating RBCs using azido sugars and click chemistry, it fundamentally builds upon well established strategies already reported by the same research group. The core methodology, systemic delivery of peracetylated azido-monosaccharides for metabolic labeling and subsequent conjugation of DBCO-bearing molecules via SPAAC, has been applied previously to dendritic cells, platelets, and adipocytes. In each case, similar labeling and in vivo targeting were made. The difference in this RBC study from previous is not the chemical mechanism, but merely the target cell type (RBCs) and the extended persistence of the azido label. While the longer half-life of labeled RBCs is notable (~42 days vs ~4 days for platelets), this reflects more a known biological property of RBCs than an engineered advance. Thus, the central novelty lies in demonstrating expected behavior in a new context. Moreover, the applications presented, such as prolonged imaging or drug delivery, are anticipated outcomes of this platform, not surprising or mechanistically new.

Response: We appreciate the Reviewer's thoughtful comment and happy to know that the Reviewer is well aware of our prior published work. As the Reviewer is quite familiar with metabolic glycan labeling, we believe the Reviewer may echo on two main challenges of this platform: i) cell-selective labeling and ii) successful labeling of some types of cells that are historically difficult to engineer and for those successful metabolic glycan labeling were deemed unlikely. For the former challenge, we spent quite some years to design trigger-activatable azido-sugars that can selectively label cells of interest under an external trigger (UV, NIR light, ultrasound, etc.) or internal trigger (hydrogen peroxide and different enzymes including histone deacetylase, cathepsin L, ALDH1A1, etc.). We have made a lot of progress

over the years, but unfortunately none of those compounds can truly ‘exclusively selectively’ label a specific type of cells. And we have to admit this path has been painful for us and others in the field. For the latter, as the Reviewer noticed, we have been working on metabolic labeling of platelets, adipocytes, and RBCs, which are all historically challenging to engineer but hold a lot of scientific, diagnostic, and therapeutic potential.

This work on RBC labeling lays out an approach to simultaneously addressing the two grand challenges in this field. On one hand, this is the first report of successful metabolic glycan labeling of RBCs either *in vitro* or *in vivo*. As RBCs are nucleus-free and do not possess the full set of protein machinery, their ability to metabolize azido-sugars and express azido-tagged glycoproteins and glycolipids on the cell membrane was deemed unlikely, if not impossible, in the field. We anticipate this manuscript will encourage more researchers to test out many other types of cells they were hesitating before. On the other hand, by leveraging the unique properties of RBCs, we show that unmodified azido-sugars (i.e., without any complexed trigger-responsive moiety) accessible to every laboratory can achieve a certain level of labeling selectivity towards the circulating RBCs in the body. This is not a trivial advance for the sake of RBC labeling/targeting itself and also an alternative strategy to achieve cell-selective metabolic labeling.

2. Although the study convincingly demonstrates that azido groups persist on the surface of circulating RBCs for over 42 days, it does not adequately establish that this labeling strategy is selective enough for practical applications in targeted imaging or drug delivery. The authors argue that azide tags decay rapidly in leukocytes and other nucleated blood cells within 3 days, thereby enabling selective RBC targeting using DBCO-functionalized molecules if administered after this window. However, this argument overlooks a more critical issue: whether normal tissues and vasculature are also being labeled and whether that labeling persists to a degree that compromises selectivity. In particular, Figure 5i-j shows detectable fluorescence signals in normal tissues (liver, spleen, etc.) after DBCO-Cy5 injection, even in the AAM-treated group. Although the signal is lower than in RBCs, it is not negligible, and there is no quantification relative to background or functional implications of this residual labeling. The authors use a “wait 3 days post AAM” strategy to reduce leukocyte labeling, but it remains unclear whether this delay also fully eliminates labeling in non-blood tissues. The Western blot and confocal data in Fig. 4i-j are helpful but limited, they do not resolve whether low-level but functionally significant tagging persists.

Response: We appreciate this another great comment from the Reviewer. We admit one key limitation of our reported RBC labeling and targeting approach is the lack of exclusive selectivity, which is and will continue to be a constant issue for any metabolic glycan labeling and targeting work. As we aimed to avoid the sophisticated design of enzyme-activatable azido-sugars and instead used the non-selective azido-sugars (AAM or AAGal), the off-target labeling of other cells in the blood and tissues is not surprising. As the Reviewer has noticed, we have shared/presented all the off-labeling labeling data in the manuscript, as part of our intention to remind our readers that the labeling of RBCs is not exclusively selective. It seems to us that the Reviewer is looking for something that can absolutely selectively label cells of interest, which is of course our and others’ ultimate goal but still has a long way to go. For now, any advance in achieving ‘more selective’ labeling and targeting of cells of interest should be encouraged and is important in this field.

Regarding Fig. 4j, we have now added the semi-quantitative data of Cy5 fluorescence intensity in different tissues as Fig. 4k, which shows negligible differences between AAM and PBS

groups in brain, heart, lung, liver, spleen, and kidney. Both the ex vivo staining data and in vivo DBCO-Cy5 conjugation data show that the azido labels in healthy tissues drops to near-background levels at 3 days or later post AAM injection. Again, we do not want to hide the fact that there would always be some levels of off-target labeling. From the pharmacology and drug development perspective, off-target labeling or binding constantly exists but should not discourage the exploration of drugs or strategies that can preferentially effect on the target cell.

Fig. 4k. Normalized Cy5 fluorescence intensity of tissues in (j).

3. Furthermore, the authors' own prior studies on metabolic labeling of platelets, adipocytes, and dendritic cells reported relatively broad azido labeling, including some tissue types. The current claim that RBCs uniquely retain azido tags while all other tissues rapidly clear them seems inconsistent with those earlier findings, suggesting either differing experimental standards or unexplored biological variability. This inconsistency undermines the claim of RBC-specific labeling and suggests that more rigorous, quantitative tissue-distribution studies are needed.

Response: We appreciate the Reviewer's mention of our previous work. Indeed, we have consistently reported that the density of azido tags on dendritic cells, platelets, and adipocytes decays rapidly to a baseline level in 2-4 days in our previous publications and in this manuscript, which is exactly the reason why the long-term labeling of RBCs is special.

4. Overall, The study is technically well-conducted and shows promise for translational applications, but it does not introduce a sufficiently novel concept or mechanism to meet the publication standards of Nature Communications. The core advances are too directly on a sequence of prior works from the same group using near-identical strategies. There is no demonstration of fundamentally new biological insight, molecular mechanism, or chemical methodology that would justify publication in a journal prioritizing broad conceptual advances. In addition, the study does not sufficiently demonstrate RBC-specific labeling or targeting, and several results raise the possibility of non-negligible off-target effects in normal tissues. This undermines the claimed value of the system for selective imaging or drug delivery. Additionally, the conclusions appear partially inconsistent with earlier work from the same group on metabolic labeling of other cell types. Without stronger and more systematic evidence of selectivity, this work does not meet the robustness or clarity needed for publication in Nature Communications.

Response: We believe these comments have been covered above. We would like to thank the Reviewer again for their deep knowledge of our prior work and excellent comments on the labeling selectivity of our approach. Novel chemistry design of unnatural sugars for cell-

selective labeling is one direction we have pursued (probably pioneered if the Reviewer agrees; [Nat Chem Biol 2017, 13, 415-424; Chem Commun 2018, 54, 4878-4881; PNAS 2023, 120, e2302342120]) for years, and we are happy to share with the community that conventional azido-sugars without any complexed chemistry design could also achieve a high level of labeling selectivity towards certain types of cells by leveraging the unique biological properties of cells. While the RBC labeling approach in this work is not exclusively selective, as expected, it provides a promising tool for researchers to explore improved RBC-based diagnostics and therapeutics.

Point-by-point reply to Reviewers' comments

(All responses were colored in blue; all changes in the manuscript were highlighted in yellow)

Reviewer #1:

During the revision process the Authors have addressed many critical points and provided new data and explaining some of the limits of this approach. This is appreciable, although not conclusive, and support acceptance from the point of view of this Reviewer.

Response: We thank the Reviewer for the positive feedback and comments, and have addressed the remaining comments in the revised manuscript.

Main section: The Authors have amended the previous version of the paper that now read "...Significant progress has indeed been made to address these issues and bring RBC-based drug delivery systems to clinical trials.^{20,21}". In addition "While direct encapsulation of cargos into RBCs has been extensively explored, most approaches rely on ex vivo engineering of RBCs,^{48,49} which inevitably inherits the common issues in RBC stability, infections, costs during the isolation, storage, ex vivo manufacturing, and infusion processes." I respectfully disagree. The technology nowadays has been greatly improved and was not only taken into phase 3 clinical trial but the trial has also documented the safety of this approach in hundreds of patients (see Ref 21). This sentence should be amended. Ref 20 is not correct since reports the title of Ref. 21

Response: We appreciate this comment from the Reviewer, and agree with the Reviewer that the past decades has witnessed significant progress in the development of *ex vivo* engineered RBC products. We have now deleted "While direct encapsulation of cargos into RBCs has been extensively explored, most approaches rely on ex vivo engineering of RBCs, which inevitably inherits the common issues in RBC stability, infections, costs during the isolation, storage, ex vivo manufacturing, and infusion processes". Deleting this sentence does not affect the flow of the manuscript at all and avoids the confusion mentioned by the Reviewer. We have also deleted Ref. 20 which was cited wrongly.

Immunogenicity: New sentence "We are aware of the concern on the potential immunogenicity of cargos being conjugated to circulating RBCs. In separate studies, we attempted to conjugate DBCO-functionalized 2,4-dinitrophenylated Hemocyanin, Keyhole Limpet (DNP-KLH) and DBCO-ovalbumin (OVA), two representative antigens, to RBCs in vivo and monitor the antibody titers over time. As a result, the in vivo conjugation of DNP-KLH and OVA antigens failed to exert any enhancement on the antibody responses compared to the control groups (Fig. S9), which lessens the concern of potent immunogenicity of RBC-conjugated cargos in our metabolic labeling approach." The new experiment is done by administering 0.1 mg of DNP-OVA or DNP-KLP that likely will partially be bound to activated RBC while partially will remain free in circulation. I'm encouraging the Authors to provide an estimate the percentage of the administered antigen that is RBC conjugated vs. the total administered.

Response: We agree with the Reviewer that only a small fraction of injected DBCO-antigens would be captured by azido-labeled RBCs. Per our estimation, likely < 1% of injected DBCO-OVA and <10% of injected DBCO-DNP-KLH would be conjugated to circulating RBCs [\sim 10% of azido-labeled RBCs, 1×10^7 RBC/ μ L, and 2 mL of circulating blood in mice yield $\sim 2 \times 10^9$ azido-labeled RBCs. Assuming $\sim 10^3$ - 10^4 azido groups per RBC based on our prior studies, circulating RBCs have a total number of 2×10^{12} - 2×10^{13} (i.e., 3.3×10^{-12} - 3.3×10^{-11} mole) azido groups. With 100% consumption of azido groups, 0.15-1.5 μ g DBCO-OVA (45 kDa) or 1.2-12 μ g DBCO-DNP-KLH (350 kDa) could be theoretically conjugated. These correspond to 0.15-1.5% I.D. for DBCO-OVA and 1.2-12% I.D. for DBCO-DNP-KLH].

We have added the following text in the revised manuscript: “We estimate < 1-10% of administered DBCO-protein could be conjugated to circulating RBCs, and the *in vivo* RBC labeling and targeting conditions require further optimization to increase the amount of conjugated cargos.”